



# Volcanic imprint in the North Atlantic climate variability as recorded by stable water isotopes of Greenland ice cores

Hera Guðlaugsdóttir[1], Jesper Sjolte[2], Árný Erla Sveinbjörnsdóttir[1] and Hans Christian Steen-Larsen[3]

[1]Nordic Volcanological Centre (NordVulk), Institute of Earth Sciences, University of Iceland
[2]Department of Geology, Lund University
[3]Geophysical Institute, University of Bergen and Bjerknes Centre for Climate Research, Bergen, Norway

*Correspondence to: Hera Guðlaugsdóttir (hera@hi.is)*

**Abstract.** Volcanic eruptions are important drivers of climate variability on both seasonal and multi-decadal time scales as a result of atmosphere-ocean coupling. While the direct response after equatorial eruptions emerges as the positive phase of the North Atlantic Oscillation in the first two years after an eruption, less is known about high latitude northern hemisphere eruptions. In this study we assess the difference between equatorial and high latitude volcanic eruptions through the reconstructed atmospheric circulation and stable water isotope records of Greenland ice cores for the last millennia (1241-1979 CE), where the coupling mechanism behind the long-term response is addressed. The atmospheric circulation is studied through the four main modes of climate variability in the North Atlantic, the Atlanti Ridge (AtR), Scandinavian Blocking (ScB) and the positive and negative phase of the North Atlantic Oscillation (NAO+/NAO-). We report a difference in the atmospheric circulation response after equatorial eruptions compared to the response after high latitude eruptions, where NAO+ and AtR seem to be more associated with equatorial eruptions while NAO- and ScB seems to follow high latitude eruptions. This response is present during the first five years and then again in years 8-12 after both equatorial and high latitude eruptions. Such a prolonged response is evidence of an ocean-atmosphere coupling that is initiated through different mechanisms, where we suspect sea ice to play a key role.

## 1 Introduction

Climate variability is mainly subjected to variations in energy and mass transfers within the Earth's atmosphere that further drive the internal dynamic forcing within the climate system. The main factors that can have a profound impact on this energy/mass transfer are anthropogenic and volcanic forcing, where the latter one is the scope of this study. Due to the physical properties of volcanic sulfate aerosols, formed when $SO_2$ from the eruptive plume reacts with water in the atmosphere, incoming short-wave solar radiation is scattered while long-wave terrestrial radiation is absorbed. For eruptions with an eruptive column reaching up into the stratosphere, this can be manifested as variations in the temperature gradients of the Earth's atmosphere, both in the vertical and latitudinal direction, that further continue to influence the atmospheric circulation and pressure systems. It is well known that



large equatorial eruptions increase the temperature gradient in the stratosphere that can lead to
amplification of the formation and transport of planetary waves (Graf et al., 1994, Kodera, 1994). In
turn, the stratospheric zonal winds are known to increase and manifest in the tropospheric climate
variability of the Northern Hemisphere as the positive phase of the North Atlantic Oscillation (NAO+),
one of the main modes of climate variability in the North Atlantic, in the first two winters after an
eruption (Robock and Mao, 1992; Graf et al., 1994, Kodera, 1994; Fisher et al., 2007; Ortega et al.,
2015). Other two modes of the North Atlantic (NA) climate variability, described through the third and
fourth EOFs of SLP (Wallace and Gutzler, 1981; Hurrell et al., 1995; Cassou et al., 2004), are the
Atlantic Ridge (AtR) and the Scandinavian Blocking (ScB) respectively. As well as having been
identified in the EOFs of SLP, these four modes have also been identified as being an important part of
the variability recorded in the isotope records of Greenland ice cores (Ortega et al., 2014). Stable water
isotopes ($^{18}$O and $^{2}$H) in the hydrological cycle preserve information regarding their transport in the
atmospheric circulation as well as other climatic parameters like temperature (Johnsen et al., 2001).
This information can both be retrieved from daily to monthly samples of precipitation as well as on a
centennial to millennial time scales from e.g. Greenland ice cores (Jouzel et al., 1997; Petit et al.,
1999). Usually when referring to stable water isotopes, the delta (δ) notation is used where the sample
value is compared to a known standard, usually Standard Mean Ocean Water (SMOW), that results in
values expressed in parts per thousand (‰).
High latitude (North Hemisphere, NH) volcanic eruptions have been less studied when it comes to the
impact on the main NA climate modes. However in a model study by Guðlaugsdóttir et al. (2018) it
was shown that NH eruptions have a slight tendency to force the atmospheric circulation towards the
negative phase of NAO (NAO-) as a result of a weakening of the stratospheric zonal winds. This
response was detected in the first 2-4 years as well as around a decade after an eruption where the study
of Guðlaugsdóttir et al. (2019) provided further evidence to support this. It has been shown that, as well
as EQ volcanic eruptions (Ottera et al., 2010; Swingedouw et al., 2015), NH volcanic eruptions too
have the ability to impact climate through the Atlantic Meridional Overturning Circulation (AMOC)
and ENSO on decadal to multi-decadal timescales (Pausata et al., 2015). However, much is still to be
learnt about the controlling mechanism behind the climate impact after NH volcanic eruptions. In this
study we compare the atmospheric winter circulation response after both EQ and NH volcanic
eruptions by analyzing 20 post-volcanic winters (referred to here as years). We assess the winter
circulation through the four main modes of NA climate variability in order to get more quantitative
information regarding the atmospheric circulation.
In this study we take advantage of the information that can be obtained from the stable water isotope
records of multiple Greenland ice cores. We hypothesize that the climate response after volcanic
eruptions can be detected in the $\delta^{18}$O records of Greenland ice cores. This involves both the direct
climate response due to changes in solar insolation as well as the indirect dynamical climate response
that indicate interactions between the main components of the climate system, the atmospheric
circulation, ocean and sea ice. We first study the volcanic response 1-5 years after each eruption in 13
shallow Greenland ice cores spanning the period 1771-1970 CE with the aim to retrieve spatial $\delta^{18}$O
pattern over the ice sheet that is associated with the atmospheric circulation response after both





equatorial and high latitude (North Hemisphere) volcanic eruptions. To compliment the information
identified in these records, the reconstructed SLP and surface temperature at 2 meters (T2m) for the
same period are also analyzed. The pattern that emerges is then tested further using longer time series,
now using three Greenland ice cores as well as the reconstruction fields, for the period of 1241-1970
CE to study 1-20 years after volcanic eruptions. A final assessment on the volcanic response identified
is then done by analyzing the average accumulation rates of three shallow North Greenland ice cores.
**2 Data & Methods**
**2.1 Greenland ice cores**
To investigate the volcanic response in $\delta^{18}O$ of Greenland ice cores, we use the winter seasonal means
(Nov-April) of 13 Greenland ice cores spanning the period 1771-1970 CE (Vinther et al., 2010). To
assess a long-term volcanic response 1-20 years after an eruption the winter seasonal mean response is
investigated in 3 Greenland ice cores, Dye 3, GRIP and Crete, spanning the period of 1241-1978 CE
(Vinther et al., 2010). Due to issues with the ice core chronology prior to ~1000 CE (Sigl et al., 2015),
and to keep consistency between the ice core and reconstruction analysis, we do not analyze volcanic
eruptions prior to 1241 CE.
**2.2 Reconstruction of the NA atmospheric winter circulation**
To further assess the spatial and temporal volcanic signal in the $\delta^{18}O$ of Greenland ice cores, both the
short and long-term volcanic signal is investigated in the NA atmospheric winter circulation
reconstructions of Sjolte et al. (2018). The reconstruction is conducted by using an isotope-enabled
version of the atmosphere-ocean model ECHAM5/MPI-OM (Werner et al., 2016) where the period
800-2000 CE is simulated. By using 8 seasonally resolved Greenland ice cores for the period 1241-
1970 CE and simulated $\delta^{18}O$, each year in the ice core data is matched with the model data by methods
described in Sjolte et al. (2018). This results in 39 significant model fits that are used to reconstruct the
atmospheric pressure, temperature at 2m (T2m) and $\delta^{18}O$ fields for the period 1241-1970. The forcing
that is implemented in the model is the same as in Jungclaus et al. (2010), except that the solar forcing
has been updated (Muscheler et al., 2016). Since our purpose is to be able to identify specific weather
regimes as a result of volcanic eruptions, it is important to have a reference. The reconstructed surface
pressure was clustered in the attempt to retrieve the four main weather regimes in the NA but without
success. Therefore the surface pressure (referred to as SLP) in ECHAM5-wiso was clustered using a k-
means clustering method where the centers were pre-defined to be 4 (k=4) according to Cassou et al.
(2004). To retrieve the most stable centroids, the calculation of cluster centroids was repeated 100
times. We also use the $\delta^{18}O$ pattern associated with the weather regimes identified in the 500mb gph of
ECHAM5-wiso that has previously been retrieved by authors (Guðlaugsdóttir et al., 2019). Although
we are not studying the 500mb gph, this provides the best estimate we have regarding the $\delta^{18}O$ pattern
that is associated with the weather regimes identifiable in the reconstructed $\delta^{18}O$ anomaly fields
($\delta^{18}O_{reconst}$). In order for a post volcanic year to be assigned with a specific weather regime, we use r
≥0.4 and p<<0.01 as a reference where the correlation tables are given in the supplementary. This
applies both for $\delta^{18}O$ and surface pressure fields.

**2.3 Extracting volcanic signal**
We select 5 equatorial (EQ) eruptions and 5 North Hemisphere (NH) eruptions to assess the volcanic
response 0-5 years after an eruption in $\delta^{18}O$ anomalies of 13 Greenland ice cores ($\delta^{18}O_{ice}$) as well as in
the reconstructed atmospheric pressure, T2m and $\delta^{18}O_{reconst}$.
The response using 3 Greenland ice cores is also assessed along with the associated reconstruction
fields with the aim to retrieve the long-term (1-20 years after an eruption). The years that are presented
in the results are selected from the average $\delta^{18}O_{ice}$ calculated from all three cores and where the
significance is by implementing a statistical Monte Carlo approach. The number of volcanic eruptions
used is presented in Table 1. The significance of the signal extracted from each core as well as
reconstructions is assessed at the 90 and 95% confidence level using a two-tailed Student's t-test, where
n is the number of volcanic eruptions stacked.

Table 1: List of all volcanic eruptions analyzed in each part. *Sigl et al. (2015)

| Eruption | Eruption year | 1241-1978 CE ($year 1 - 5$) (n=8) | 1241-1978 CE ($year 8 - 20$) (n=8) | 1771-1970 CE ($year 0 - 5$) (n=5) | 1724-2011 CE ($NEEM$) (n=6) | Forcing* ($W/m^2$) |
|---|---|---|---|---|---|---|
| **Equatorial** | | | | | | |
| Samalas | 1258 | x | x | | | -32.8 |
| Unknown | 1276 | x | x | | | -7.7 |
| Unknown | 1345 | x | x | | | -9.4 |
| Kuawe | 1458 | x | x | | | -20.1 |
| Huyaputina | 1601 | x | x | | | -11.6 |
| Parker | 1641 | x | x | | | -11.8 |
| Banda Api | 1695 | x | x | | | -10.2 |
| Unknown | 1809 | | | x | x | -12.0 |
| Tambora | 1815 | x | x | x | | -17.2 |
| Cosiguina | 1836 | | | x | x | -6.6 |
| Krakataua | 1884 | | | x | x | -5.5 |
| Agung | 1963 | | | x | x | -3.8 (1964) |
| El Chichon | 1982 | | | | x | -0.4 |
| Pinatubo | 1991 | | | | x | -6.5 (1992) |
| **NH** | | (n=8) | (n=7) | (n=5) | (n=6) | |
| Hekla | 1300 | x | x | | | – |
| Öræfajökull | 1362 | x | x | | | – |
| Veidivötn | 1477 | x | x | | | -3.1 |
| Katla | 1721 | x | x | | | -0.8 |
| Katla | 1755 | x | x | | x | -0.9 (1756) |
| Hekla | 1766 | | | | x | -1.4 |
| Laki | 1783 | x | x | x | x | -15.5 |
| Askja | 1875 | x | | x | x | -0.6 |
| Novarupta | 1912 | | | x | | -3.3 |
| Katla | 1918 | | | x | | -0.7 (1919) |
| Hekla | 1947 | x | x | x | x | -0.6 |
| Hekla | 1970 | | | | x | – |


The EQ eruptions are selected based on Sigl et al. (2015) where the individual eruptions selected do no
interfere with one anther within the time frame analyzed (either 1-5 or 1-20 years). The radiative
forcing for the EQ eruptions is between -3.8W/m² and -32.8W/m² with the exception of El Chichon
that is -0.4W/m². However since it is considered to be one of the largest eruptions of the 20th century it
is included. It can be difficult to select an ideal candidate for the analysis of the climate response after
NH eruptions since they do occur frequently but few have the ability to alter the climate system. Many



of the NH volcanic eruptions that have been selected in Table 1 are considered to be one of the largest
volcanic eruptions in the last millennium, where all but one NH eruption listed in the table is Icelandic.
Some of these eruptions erupted for a year or more (Hekla 1300 and 1766) others lasted a little less
than a year (Laki 1783 and Askja 1875) although the duration of the majority of the eruptions is 1-5
months (Janebo et al., 2016; Carey et al., 2009; Thordarson et al., 2003; Thordarson and Larsen, 2007;
Sharma et al., 2008). We also use the reconstructed forcing by Sigl et al. (2015) to select these NH
eruptions except for two events, the Hekla 1300 and Öræfajökull 1362. Hekla 1300 CE lasted a year
and has been assigned a VEI index of 4 while Öræfajökull erupted for several months (June-Oct) with a
VEI index of 5-6.
**3 Results**
**3.1 Equatorial volcanic response**
**3.1.1 Atmospheric circulation response 0-5 years after EQ eruptions: 1771-1970 CE**
Figures 1-3 show results for the atmospheric circulation response 0-5 years (winters) after EQ volcanic
eruptions. Since year 0 is the year of the volcanic eruptions, it includes little volcanic perturbations but
is kept for comparison. It is however displayed for comparison. The volcanic eruptions were selected
according to Table 1, where 5 volcanic eruptions were stacked in $\delta^{18}O_{ice}$ of 13 cores, as well as
$\delta^{18}O_{reconst}$ and surface pressure reconstructions, to retrieve a composite anomaly signal calculated with
respect to 10 years prior to each event. See T2m post volcanic fields in supplementary Figure F2. To
assess the post-volcanic reconstruction fields, we use the average $\delta^{18}O$ and surface pressure pattern
(supplementary Figure F1) associated with each weather regime that has been retrieved from
ECHAM5-wiso (supplementary Table T1-T2).





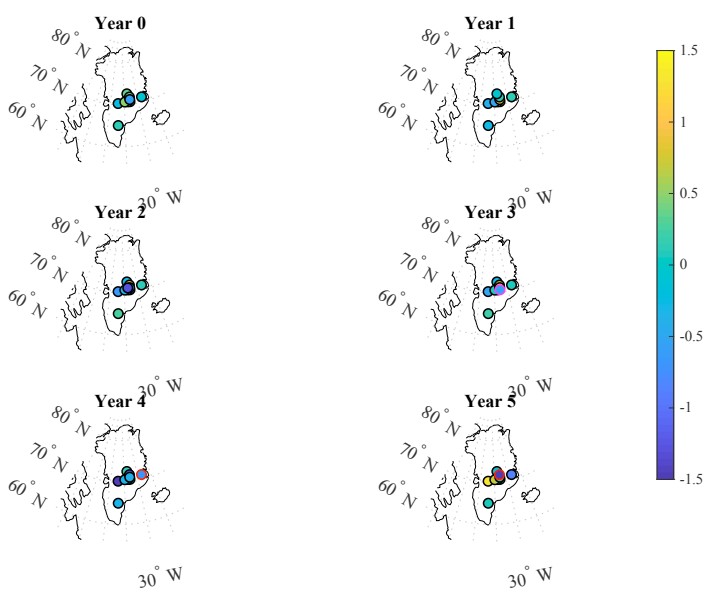


Figure 1: Composite $\delta^{18}O_{ice}$ anomalies of 13 Greenland ice cores (1771-1970 CE) 0-5 years after 5

equatorial volcanic eruptions. Magenta circles (years 1 and 2) are sites with significance at the 90%

confidence level and red circles (years 4 and 5) are sites with significance at the 95% confidence level.


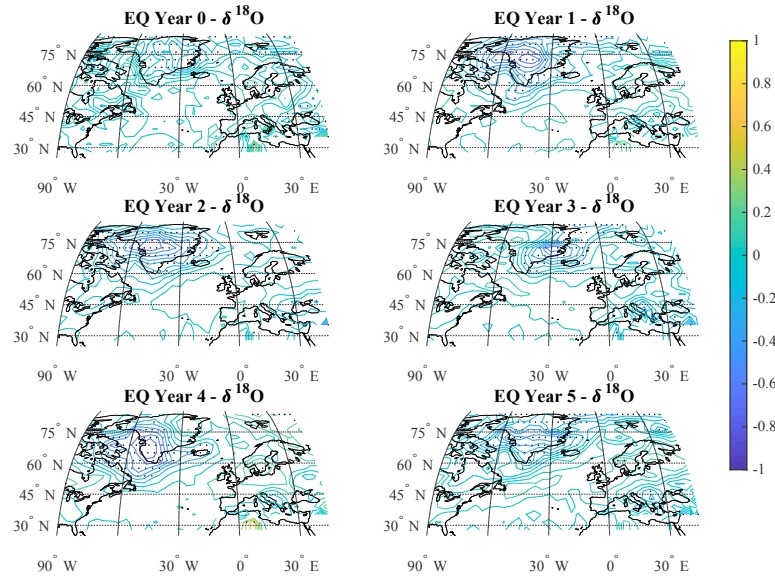


Figure 2: The short-term $\delta^{18}O_{reconst}$ response 0-5 years after 5 EQ volcanic eruptions.






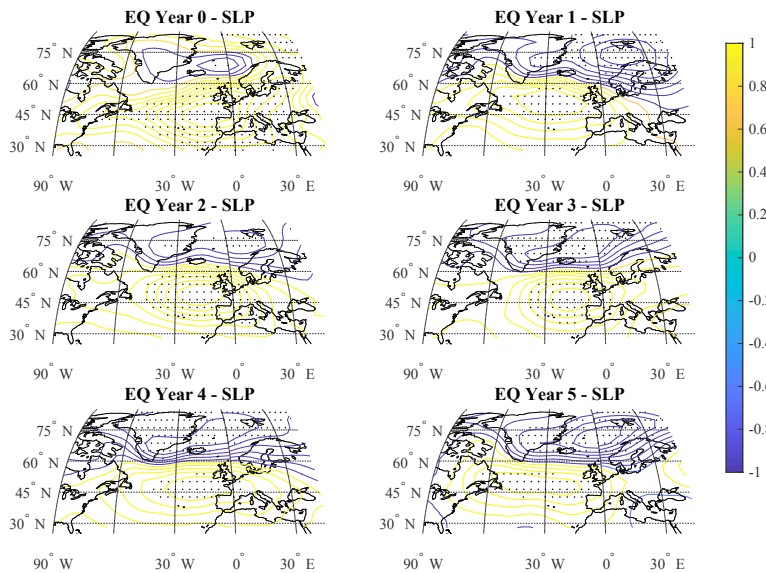


Figure 3: The short-term surface pressure response 0-5 years after 5 EQ volcanic eruptions.

A general agreement can be found between the spatial $\delta^{18}O_{ice}$ pattern identified in Figure 1 and the
$\delta^{18}O_{reconst}$ pattern over Greenland in Figure 2 for all years with year 5 being an exception. A clear
atmospheric circulation response emerges in years 1-5 in all parameters analyzed, where $\delta^{18}O_{ice}$
becomes significant in year 3 at the 90% c.l. and year 4-5 at the 95% c.l. The $\delta^{18}O_{ice}$ anomaly pattern in
years 3-4 is on average negative while a gradient is present in year 5 with more positive anomalies in
SW-Greenland that become more negative towards NE. When compared with the $\delta^{18}O$ pattern
associated with each weather regime, this indicates the presence of NAO+. While $\delta^{18}O_{reconst}$ is less clear
on this, with a weak correlation with AtR in year 3 (r=0.37) while years 1-2 and 4-5 correlates with
NAO+ (r=0.49-0.67, p<0.01), the SLP shows a clear NAO+ pattern present in all years 1-5 (r>0.7 and
p<0.01). This is also detected in the T2m (supplementary Figure F2).

**3.1.2 Atmospheric circulation response 1-4, 8-11 and 17-20 years after EQ eruptions: 1241-1978**
**CE.**
Figures 4-6 show results for the atmospheric circulation response where the significant years detected
in the 3-core average (Figure 4a) have been retrieved from the reconstruction fields and their anomalies
calculated with respect to 10 years prior each event. The volcanic eruptions were selected according to
Table 1, where 8 volcanic eruptions were stacked in $\delta^{18}O$ of 3 cores ($\delta^{18}O_{ice}$), as well as in the $\delta^{18}O_{reconst}$
and SLP fields to retrieve a composite anomaly signal 1-4, 8-11 and 17-20 years after EQ eruptions.
See T2m post volcanic fields in supplementary Figure F3. To assess the post-volcanic reconstruction
fields, we use the average $\delta^{18}O$ and surface pressure pattern that is associated with each weather regime
(supplementary Table T1-T2).





a)

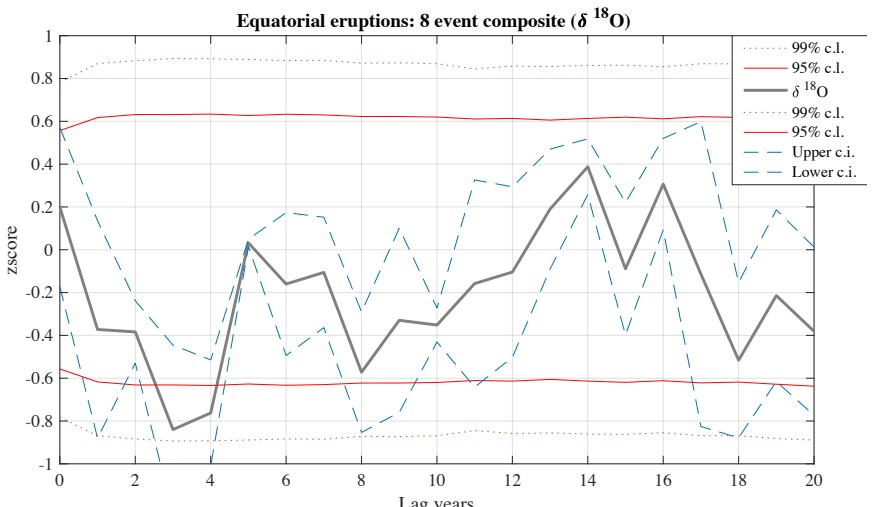

b)

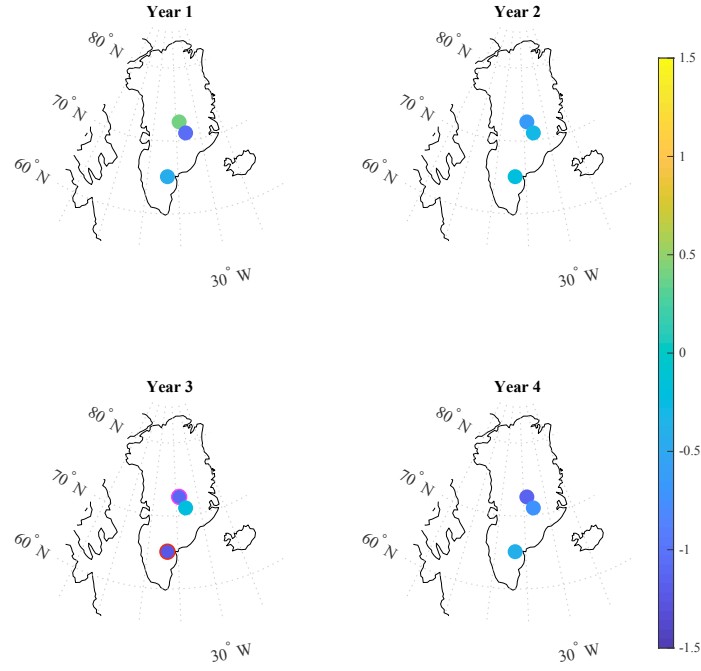

c)



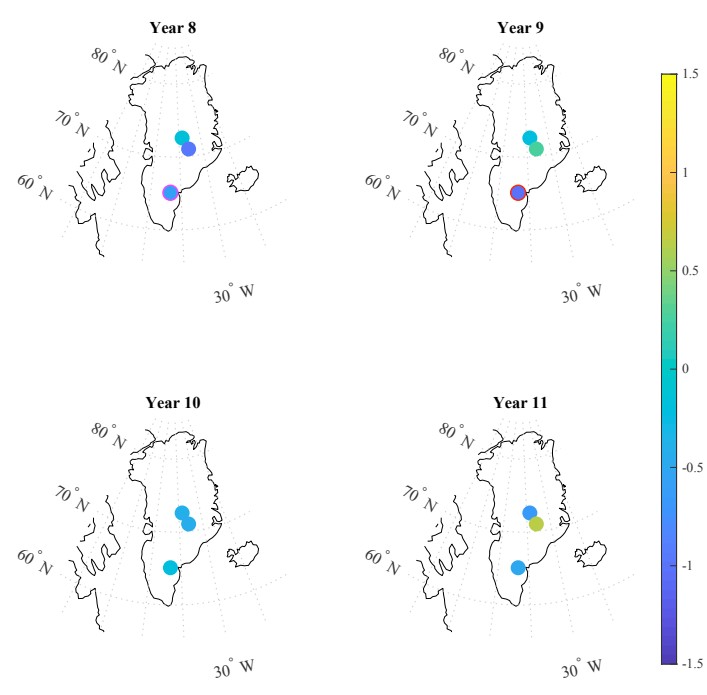

d)





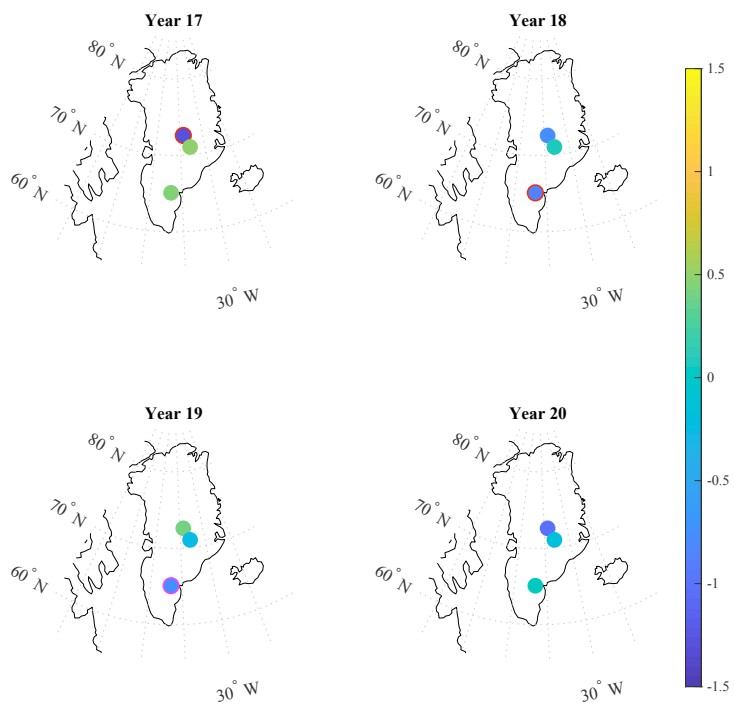


Figure 4: a) The average δ^18O of all three ice cores (Dye 3, GRIP and Crete). Based on significant years
in a) the δ^18O response in years 1-4 (b), years 8-11 (c) and years 17-20 (d) is shown where number of
EQ volcanic eruptions in the period of 1241-1978 CE is 8.





a)

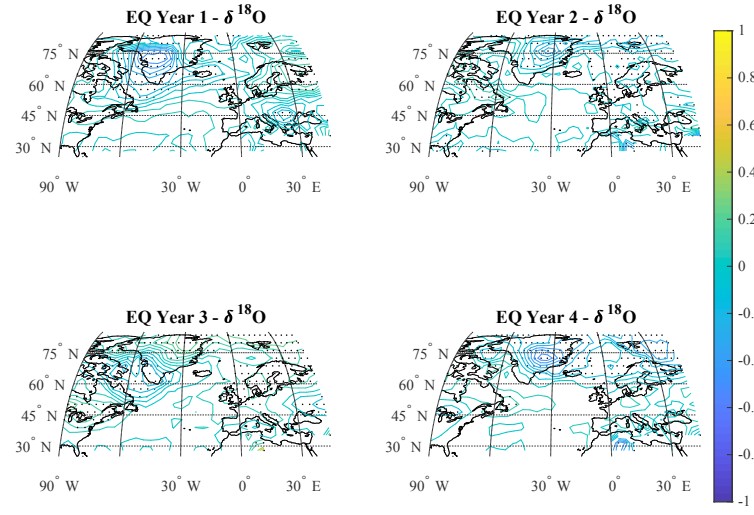

b)

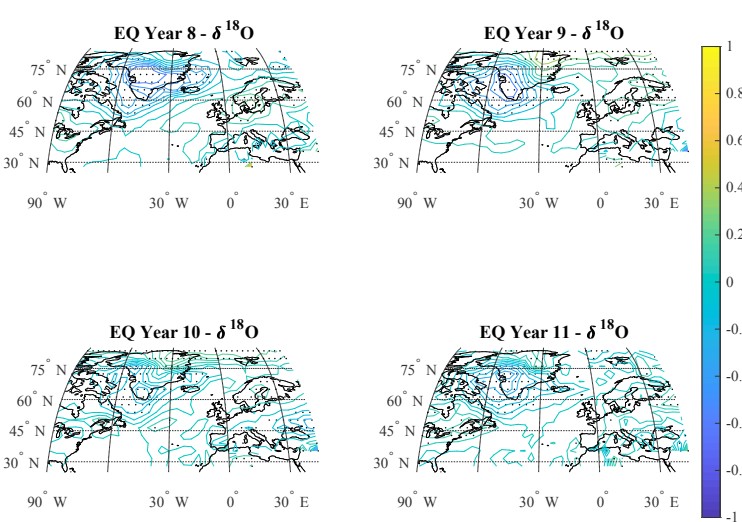






c)

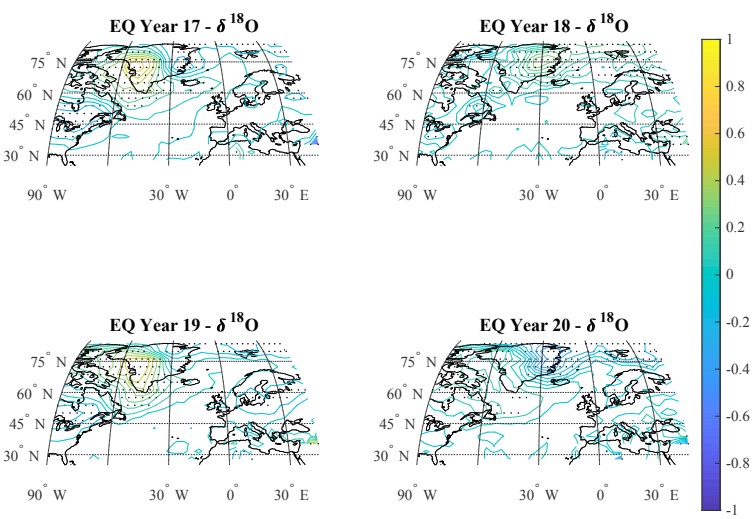


Figure 5: The composite $\delta^{18}O_{reconst}$ response a) 1-4 years, b) 8-11 years and c) 17-20 years after 8 EQ
volcanic eruptions.

a)

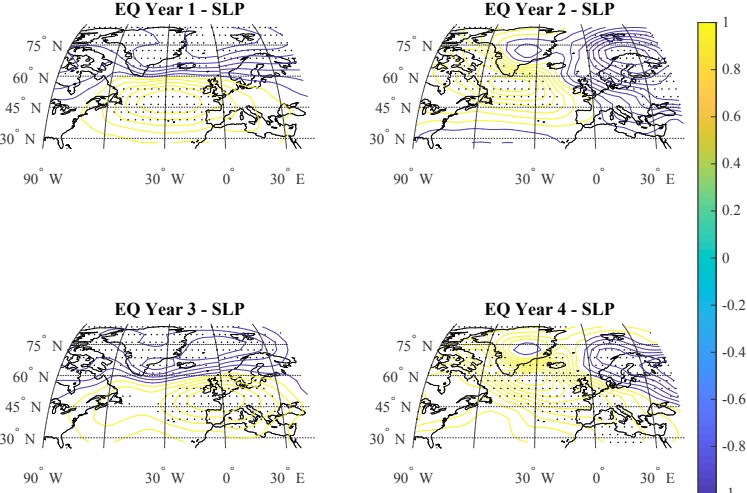












b)

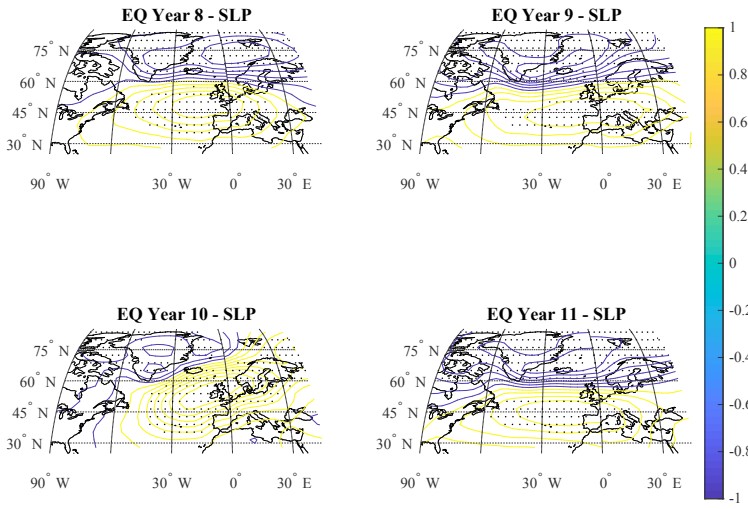


c)

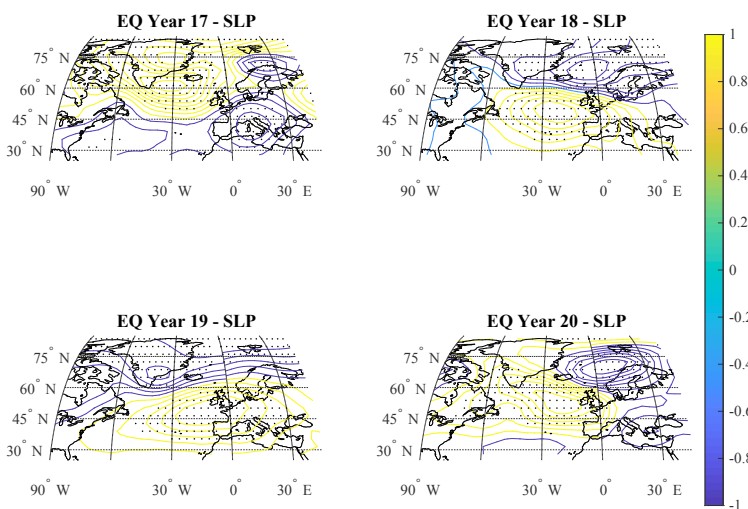



Figure 6: The composite SLP response a) 1-4 years, b) 8-11 years and c) 17-20 years after 8 EQ
volcanic eruptions.

The $\delta^{18}O_{ice}$ pattern in Figure 4 is consistent with the $\delta^{18}O_{reconst}$ pattern for all years except year 3 where
positive anomalies are present while $\delta^{18}O_{ice}$ show negative anomalies that are significant in the 95%
(Dye 3) and 90% (GRIP) c.l. A persistent NAO+ pattern is present in both $\delta^{18}O_{ice}$ and SLP in years 8-
11 in Figure 5b and 6b except for ScB being present in reconstructed $\delta^{18}O$ in year 10 (r=0.57).
Negative anomalies are present in $\delta^{18}O_{ice}$, of which year 8 is significant in the 90% c.l. and year 9 in the



95% c.l. that strongly supports such a prolonged NAO+ signal. This is further supported in the T2m
fields (Figure F3).
Figure 5 shows a clear NAO+ response in $\delta^{18}O_{reconst}$ in years 1 (r=0.54) and 3 (r=0.53) and an AtR in
year 2 (r=0.60) and 4 (r=0.64) that is supported by the SLP in Figure 6 as well as in the T2m fields of
Figure F3. During an NAO+, negative anomalies are to be expected over Greenland. Since this pattern
is present in all years of Figure 5a and 6a, it is difficult to distinguish between NAO+ and AtR in the
ice cores (Figure 4a) since more spatial data would be required to identify AtR.
Years 17-20 (Figure 5c and 6c) do not show as much consistency between $\delta^{18}O_{reconst}$ and SLP
compared to prior years analyzed. However, year 17 shows a clear NAO- in both $\delta^{18}O_{reconst}$ (r=0.71)
and SLP (r=0.75). $\delta^{18}O_{ice}$ of year 17 is in support of an NAO- being present where $\delta^{18}O$ of GRIP result
in a 95% significant negative anomaly while the other two are slightly more positive (although
insignificant). The same site is also significant at the 95% c.l. (with negative anomaly) in year 18 but
this is not in agreement with the ScB pattern that is present in $\delta^{18}O_{reconst}$ (r=0.55), showing positive
anomalies being present over Greenland ice sheet, also confirmed in the T2m fields (Figure F3).

**3.2 North Hemisphere volcanic response**

**3.2.1 Atmospheric circulation response 0-5 years after NH eruptions: 1771-1970 CE**
Figures 7-9 show the atmospheric circulation response after 5 composite North Hemisphere (NH)
volcanic eruptions where the anomalies are calculated with respect to 10 years prior to each event. As
for the EQ eruptions, year 0 is the eruption year and thus has little volcanic perturbations but is kept
here for comparison. The volcanic eruptions were selected according to Table 1, where the volcanic
eruptions were stacked in $\delta^{18}O$ of 13 cores ($\delta^{18}O_{ice}$), as well as $\delta^{18}O_{reconst}$ and SLP field reconstructions
to retrieve a composite anomaly signal 0-5 years after NH eruptions. T2m fields can be found in
supplementary Figure F6. As for the EQ eruptions, the post-volcanic reconstruction fields are assessed
by using the average $\delta^{18}O$ and surface pressure pattern associated with each weather regime
(supplementary Table T3-T4).



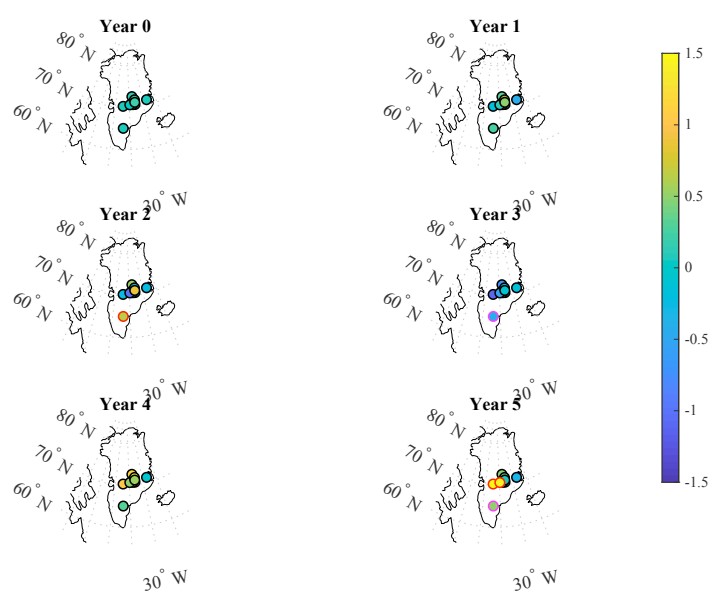


Figure 7: Composite δ¹⁸O anomalies of 13 Greenland ice cores (1771-1970 CE) 0-5 years after 5 North
Hemisphere volcanic eruptions.


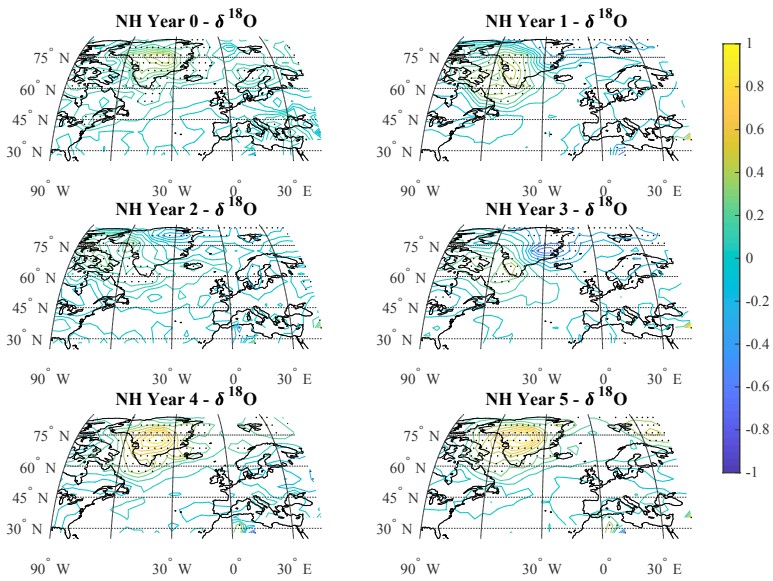


Figure 8: The short-term δ¹⁸O response 0-5 years after 5 NH volcanic eruptions.



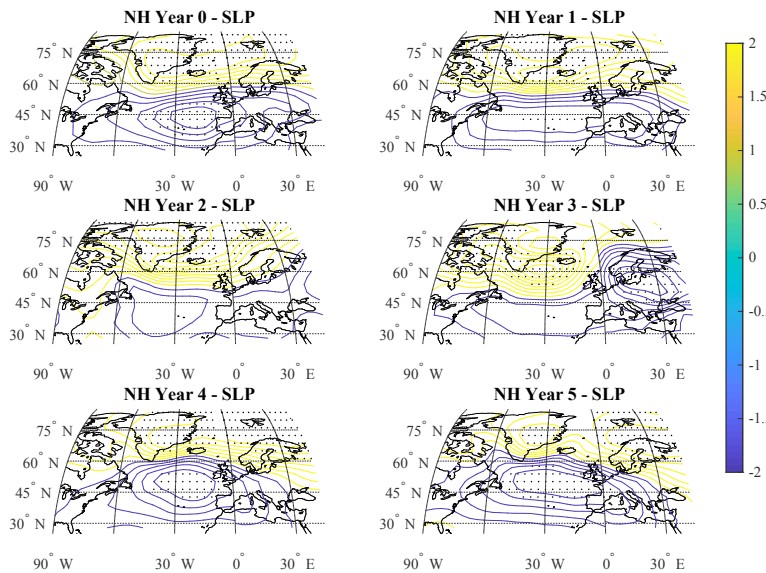


Figure 9: The surface pressure response 0-5 years after 5 NH volcanic eruptions.


The pattern emerging in the $\delta^{18}O_{ice}$ (Figure 7) is in agreement with $\delta^{18}O_{reconst}$ (Figure 8) where negative
anomalies develop in years 1-3 followed by positive anomalies in years 4-5. $\delta^{18}O_{ice}$ is consistent with
$\delta^{18}O_{reconst}$ and SLP in years 1 and 2 where a clear NAO- emerges ($\delta^{18}O_{ice}$: r=0.65 and 0.66 respectively,
SLP: r=0.96 and 0.89 respectively). In year 3, AtR is present in both fields (r=0.67 and r=0.71). This is
consistent with the negative anomalies detected in the $\delta^{18}O_{ice}$ in years 1-3 and become significant in
years 2 (95% c.l.) and year 3 (90% c.l.). In years 4-5 clear positive anomalies emerges in $\delta^{18}O_{ice}$ that is
expected during an NAO+, also displayed in $\delta^{18}O$, while NAO- is present in SLP. The $\delta^{18}O_{ice}$ for years
4 and 5 show positive anomalies at almost all sites that become significant at the 95% c.l. at two sites
in year 5 (Figure 4) and 90% significant at Dye 3. These positive anomalies are also captured in $\delta^{18}O$
and T2m (Figures 5 and Figure F4) where the pattern correlates weakly to both ScB and NAO-.

**3.2.2 Atmospheric circulation response 2-5, 8-10 and 16 years after NH eruptions: 1241-1978 CE**
Figures 10-12 show results for the atmospheric circulation response where the significant years
detected in the 3-core average (Figure 4a) have been retrieved from the reconstruction fields.
Anomalies are calculated with respect to 10 years prior to each event. The volcanic eruptions were
selected according to Table 1 where volcanic eruptions were stacked to retrieve the volcanic signal in
$\delta^{18}O_{ice}$ of 3 cores, as well as $\delta^{18}O_{reconst}$ and SLP, to retrieve a composite anomaly signal 2-5, 8-10 and
16 years after NH eruptions. To avoid mixing with large EQ eruptions, only 7 NH volcanic eruptions
are stacked to retrieve the long-term response compared to 8 volcanic eruptions that are stacked to
retrieve the short-term response. See T2m post volcanic fields in supplementary Figure F5 and
supplementary Table T3-T4.






a)

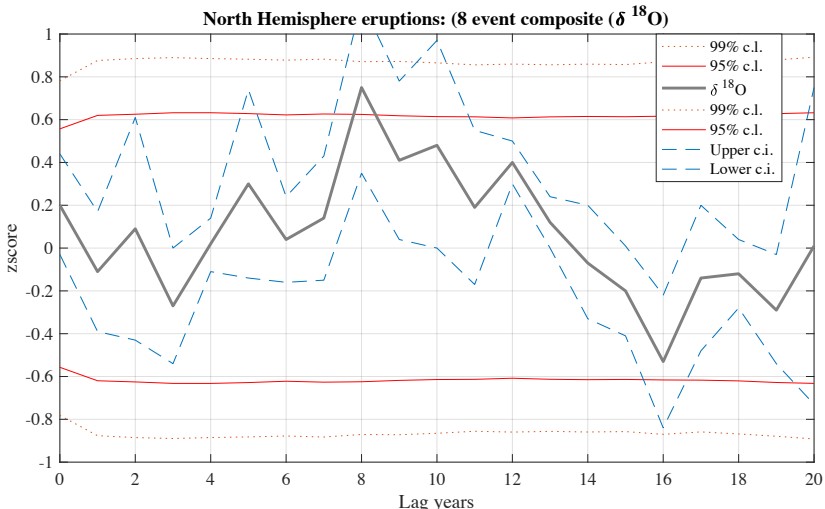


b)

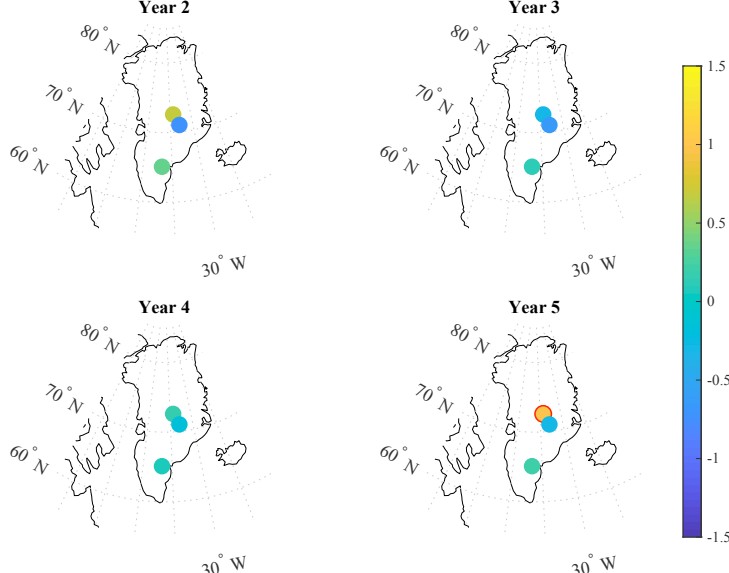






c)

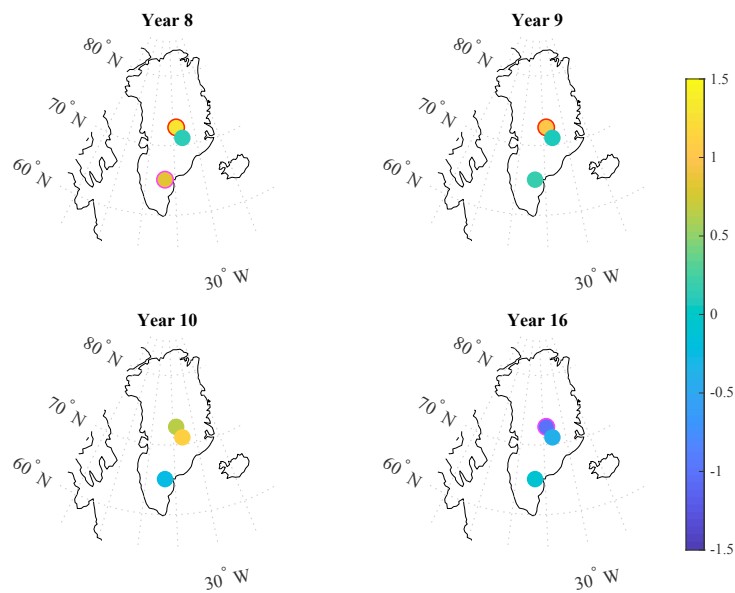

Figure 10: a) The short-term and b) long-term $\delta^{18}O$ response after 8 and 7 NH volcanic eruptions
respectively, retrieved by stacking 3 Greenland ice cores spanning 1241-1978 CE. In c) the $\delta^{18}O$
retrieved by averaging Dye 3, GRIP and Crete is illustrated, where the red circle refers to a year of 95%
significance according to a Student's t-test and the cyan circles are 90% significant.
a)

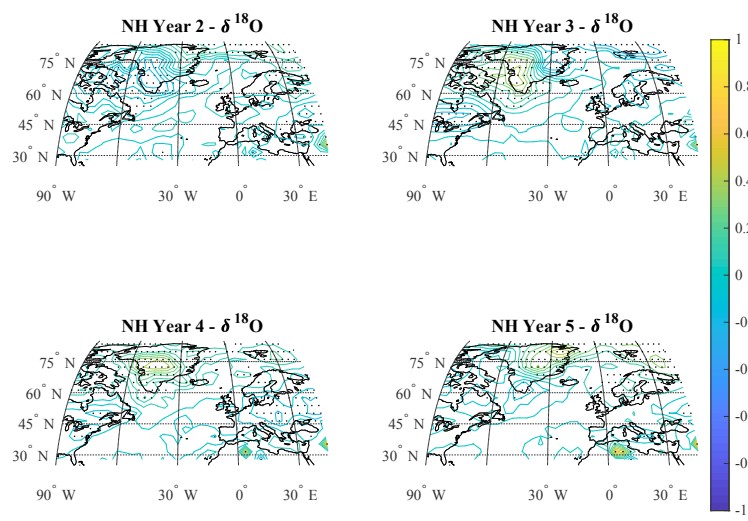







b)

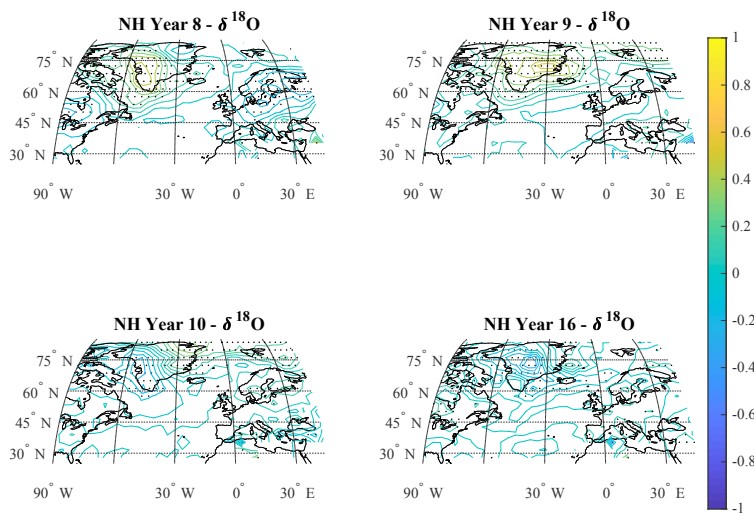


Figure 11: The reconstructed δ18O fields a) 2-5 years and b) 8-10 and 16 years after 8 and 7 NH
volcanic eruptions, respectively.

a)

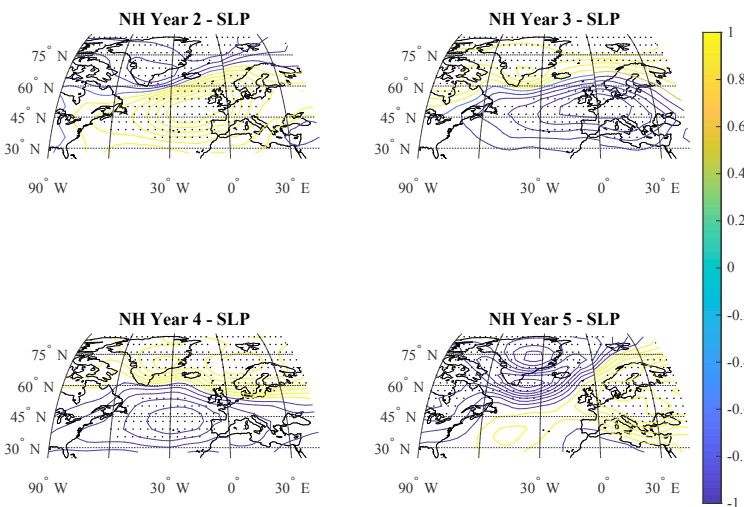










b)

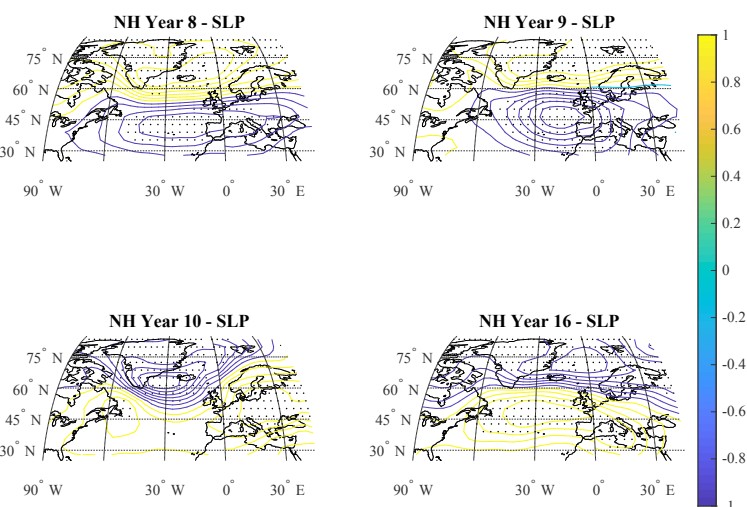


Figure 12: The reconstructed $\delta^{18}$O fields a) 2-5 years and b) 8-10 and 16 years after 8 and 7 NH
volcanic eruptions, respectively.

The spatial $\delta^{18}$O$_{ice}$ pattern in Figure 10 shows a consistent relationship with the pattern emerging over
Greenland in $\delta^{18}$O$_{reconst}$ (Figure 11). In Figure 10a the atmospheric circulation response that emerges in
the $\delta^{18}$O$_{ice}$ anomalies is not as clear as compared to Figure 7, where year 5 is only significant at the
95% c.l. In $\delta^{18}$O$_{reconst}$ and SLP fields (Figures 11a and 12a respectively), using a composite of 8
eruptions, ScB emerges in year 2 (r=0.57) while NAO+ emerges in the SLP (r=0.78). NAO- emerges in
both the $\delta^{18}$O and SLP (r=0.86 and r=0.68 respectively) in year 3 (r=0.86 and r=0.56 respectively) and
4 (r=0.56 and r=0.86 respectively). Similar to year 2, ScB emerges in the reconstructed $\delta^{18}$O (r=0.69) in
year 5 while NAO+ is present in the SLP (r=0.60). The T2m fields provide a support for the $\delta^{18}$O$_{reconst}$,
suggesting that ScB and NAO- are dominating in the first 5 years after NH eruptions.

Years 8 and 9 (Figure 11b and 12b) show $\delta^{18}$O$_{ice}$ anomalies significant in the 95% c.l. being present at
GRIP as well as 90% significant at Dye 3 in year 8. These positive anomalies of year 8 are associated
with a clear NAO- being present in both $\delta^{18}$O (r=0.84) and SLP (r=0.95), also present in the
reconstructions of year 9 (r=0.49 and r=0.79 respectively). As for year 2 and 5, year 10 has an ScB
present in $\delta^{18}$O$_{reconst}$ (r=0.63) while NAO+ is present in SLP (r=0.76). On average, the SLP fields detect
NAO+ or NAO- more frequent compared to ScB or AtR. This is not the case for the other fields
($\delta^{18}$O$_{reconst}$ and T2m) and therefore the SLP is perhaps more sensitive towards NAO compared to the
other regimes. A clear NAO+ emerges in year 16 in the reconstructions ($\delta^{18}$O: r=0.43, SLP: r=0.90) as
well as significant negative anomalies being present in $\delta^{18}$O$_{ice}$, a further evidence for the presence of
NAO+.

395

396



## 4 Discussions & conclusions

The results show a consistent atmospheric circulation response in years 1-4 after EQ volcanic eruptions in both the reconstructions of Sjolte et al. (2018) and $\delta^{18}$O of Greenland ice cores, where both AtR and NAO+ emerge in the first four years after an eruption. An increase in the frequency of AtR was also detected in year 2 after EQ eruptions using six ECHAM5 ensemble members (including ECHAM5-wiso) in Guðlaugsdóttir et al. (2018) where the authors concluded that the signal emerges as a result of volcanic surface cooling. According to ECHAM5-wiso, a $\delta^{18}$O gradient is present over the Greenland during an AtR with negative anomalies present in the NE part of Greenland that becomes positive towards Baffin Island in the SW. The opposite occurs during NAO+, where more positive anomalies are present in the NE while more negative anomalies are present in the SW where the gradient (the $\delta^{18}$O difference between NE and SW) is stronger during NAO+. This difference is not clear when it comes to the identification of AtR in $\delta^{18}O_{ice}$, where more data from sites further north would benefit the identification agreeing with Guðlaugsdóttir et al. (2019). The significance of the negative $\delta^{18}O_{ice}$ anomalies of years 3-5 using a 5 EQ eruption composite, along with results from the reconstructions (Figure 1-3), clearly indicate an NAO+ that is more prolonged than the expected NAO+ signal in the first 2 years (Robock and Mao, 1992; Graf et al., 1994; Kodera, 1994; Fisher et al., 2007; Ortega et al., 2015). Although this NAO+ signal is not as clear in the $\delta^{18}O_{ice}$ using 8 a composite of EQ eruption, $\delta^{18}O_{ice}$ becomes significant in year 3 where all years show negative anomalies. However, as mentioned, this does not rule out the presence of AtR in $\delta^{18}O_{ice}$. This prolonged NAO+ signal was previously reported by Sjolte et al. (2018) where they identified an increase in the NAO index in the first five years after EQ eruptions. Therefore similar results should be assumed. Furthermore, in Guðlaugsdóttir et al. (2018) an increase in the frequency of NAO+ was identified 3-5 years after EQ eruptions. It has been suggested that the volcanic surface cooling is overestimated in ECHAM5 (Driscoll et al., 2012) where the model has failed to produce an NAO+ in the first two years after EQ eruptions, as would otherwise be expected. However, an NAO+ is identified in the reconstructions in years 1-5 and therefore they do not seem to depend on this volcanic sensitivity that seem to be present in ECHAM5. To answer if this AtR response in years 2 and 4 is indeed a natural response to large EQ eruptions, more data and analysis is required. The identification of this prolonged NAO+ pattern in $\delta^{18}O_{ice}$ is important in order to establish a mechanism behind such a delayed response. Both the $\delta^{18}$O and T2m in Figures 2a, 5a and F2 and F3 provide us with evidence of a temperature decrease in the Nordic Seas that would again be in support of the known sea ice increase after EQ eruptions in that region. This sea ice increase could lead to NAO+ via an increase in planetary wave formation and transport, where it is known that the opposite occurs during a sea ice decrease leading to NAO- (Kim et al., 2014; Magnúsdóttir et al., 2004). In our case it seems that although an increased latitudinal temperature difference in the stratosphere initially leads to NAO+ in year 1, the NAO+ remains to be persistent as a result of an increase in planetary wave formation. However, the AtR also participates in this climate response as a result of surface cooling. The strength of this prolonged NAO+ response seems to depend on the number of volcanic eruptions being stacked, where the 5 EQ eruptions composite results in a stronger and more persistent NAO+ signal (years 1-2 and 4-5) compared to the 8 EQ eruption composite (years 1 and 3). AtR emerges using both 5 (year 3) and 8 composite EQ eruptions (2 and 4).





The volcanic signal 8-11 years after EQ eruption is clear in $\delta^{18}O_{ice}$ where negative anomalies are in
support of the persistent NAO+ present in the reconstruction fields. The significance of year 10 when
the cores are averaged also provides a support for the detected NAO+ (Figure 6c). Zanchettin et al.
(2012) identified a similar volcanic response as a winter warming, where NAO+ emerged in years 10-
12, while Guðlaugsdóttir et al. (2018) identified the emergence of NAO+ two years later. This timing
also agrees with the known decadal increase in Atlantic Meridional Overturning Circulation (AMOC)
identified in Swingedouw et al. (2015). The response detected in years 17-20 is not as robust compared
to previous years analyzed. However, a robust NAO- response emerges in year 17 as a significant
negative $\delta^{18}O_{ice}$ anomaly at GRIP and in the pattern detected in reanalysis fields. This is followed by a
significant negative $\delta^{18}O_{ice}$ anomaly at Dye 3 that is evidence for the presence of NAO+ as is detected
in SLP in year 18 and 19. According to Swingedouw et al. (2015), an increase in North Atlantic SST
(25-55°N) reaches a maximum around year 20, occurring at the time of a slight weakening detected in
the AMOC. We do not see evidence for this SST increase in our $\delta^{18}O$ and T2m fields although the
presence of NAO+ in $\delta^{18}O_{ice}$ and SLP does indicate a warming pattern over the Northern Hemisphere
landmass. However, the causality between our results and a weakening of the ocean gyre circulation
(AMOC) can only be speculative at this stage.

The atmospheric circulation response 0-5 years after NH eruptions according to our results emerges as
a robust NAO- response in years 1 and 2 by using 5 NH eruption composite while NAO- emerges in
year 3 and 4 using 8 eruption composite. In both year 2 and 5, NAO+ is detected in the SLP but ScB in
the $\delta^{18}O$. While the reconstructed SLP fields result in an NAO+ or NAO- - like pattern for almost all
years analyzed for both NH and EQ eruptions (an exception being year 2, 4 and 20 after EQ eruptions),
the same is not to be said regarding the reconstructed $\delta^{18}O$ and T2m fields. When the clustering
method, also used on the SLP of ECHAM5-wiso (K-means clustering, k=4), is used on the
reconstructed SLP only NAO emerges (not shown). It therefore seems that years when ScB emerges in
the $\delta^{18}O$ the SLP detects it as NAO+ (year 2 and 5 after NH eruptions as well as year 18 after EQ
eruptions), since the SLP has a tendency towards NAO although it is able to detect AtR. When the
$\delta^{18}O_{ice}$ is considered, both for 5 and 8 NH eruption composite, our results are more in favor of ScB,
especially for year 5. This is in agreement with Guðlaugsdóttir et al. (2018) that suggests that weather
regimes that are associated with a weaker stratospheric polar vortex, like AtR, ScB and NAO-, are
more common in the first years after NH volcanic eruptions compared to NAO+ that is evident of a
stronger polar vortex. It was first proposed by Graf et al. (1994) that NH eruptions could weaken the
stratospheric polar vortex due to the decrease in meridional (latitudinal) temperature gradient. The
identification of weather regimes associated with such a polar vortex weakening provides an
opportunity to forecast potential extreme events in the aftermath of NH eruptions, where it can be
assumed based on our results that such eruptions do not need to be as large as EQ eruptions to have an
impact. The long-term atmospheric circulation response after NH eruptions according to our results is a
robust NAO- in years 8 and 9. Again, the SLP detects NAO+ in year 10 while $\delta^{18}O$ detects ScB. When
the $\delta^{18}O_{ice}$ is considered, the pattern emerging in year 10 is more in favor of ScB but this pattern is not
significant. Year 16 results in a robust NAO+ pattern in both the $\delta^{18}O_{ice}$ and the reconstruction fields.



The key to answer questions regarding the long-term (decadal/multi-decadal) response lie within the
ocean and sea ice response to volcanic eruptions, and since these components are outside of the scope
of this study we can only speculate about the possible mechanism. In a study by Pausata et al. (2015) a
sudden decrease in the Ocean Heat Content was reported in the first 5 years due to the volcanic surface
cooling after high latitude eruptions. The OHC increased slowly until year 10-15 where it reached a
steady state (below average) throughout the study period. This offers one explanation to the emerging
NAO+ in year 16 after NH eruptions, where the OHC increase result in an NAO+ -like SST pattern. It
can be observed in the reconstructed $\delta^{18}$O fields that the area over Greenland is significant in all years
analyzed, both for EQ and NH eruptions (Figures 2, 5, 8 and 11), indicating a bias in the
reconstructions towards Greenland since it is less significant towards the northern mid latitudes.
However, it is unlikely that this bias would favor any particular weather regime although the centers of
action in the weather regimes may show tendencies towards the Greenland ice sheet that would
otherwise vary.
The pattern that appears to be emerging here is a difference in the atmospheric circulation response
after EQ eruptions compared to the response after NH eruptions, where NAO+ and AtR seems to be
more associated with EQ eruptions while NAO- and ScB seems to follow NH eruptions. To assess this
anti-phase in the response of EQ and NH eruptions, we move further north over the Greenland ice sheet
to study the accumulation in the shallow ice cores of NEEM and test if a difference in accumulation
can be detected. Although ~70% of the accumulation at NEEM falls during summer, NEEM offers the
possibility to examine this further. When the basic definition of the different phases of NAO is
considered in terms of precipitation amount, NAO+ is on average associated with less precipitation
over Greenland while NAO- is associated with more precipitation (Hurrel, 1995). Our results, done by
stacking six EQ and NH volcanic eruptions (see Table 1) in similar manner as above, are presented in
Figure 13 where this relationship is only weakly indicated. The accumulation decreases slightly in
years 1-4 where the NAO index (gray and blue dashed lines) remains below the 95% threshold,
indicating a slight accumulation decrease. A slight increase is observed in the trend of the lower c.i.
(blue dashed line) after NH eruptions in years 1-4 where the upper 95% c.i. remains above the 95%
threshold. However these changes are weak in our results and the question regarding precipitation
changes after both EQ and NH volcanic eruption and if it can be associated with the two different
phases of NAO (as well as AtR and ScB) demands more data before it can be answered.

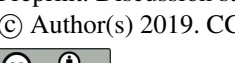


a)

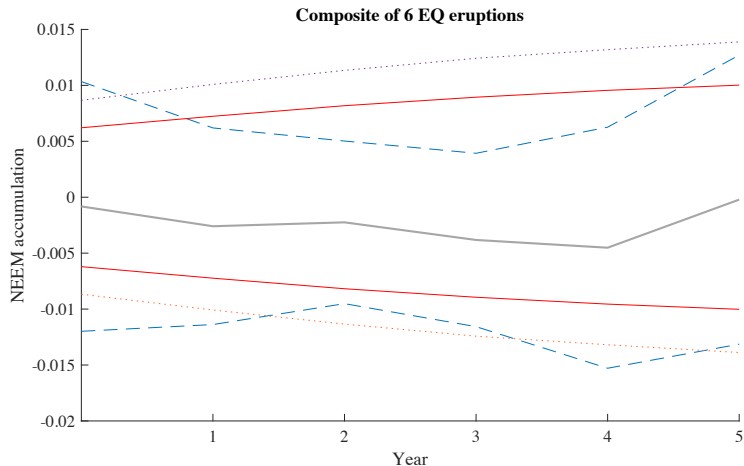

b)

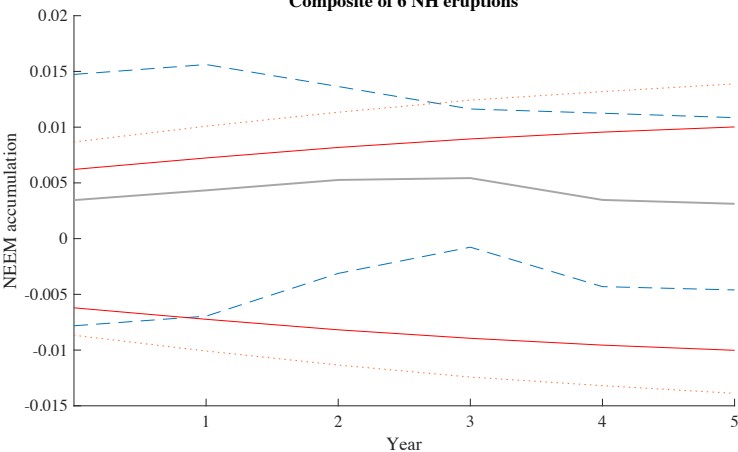

Figure 13: The composite volcanic response in the accumulation at NEEM 0-5 years after an eruption
using a) 6 EQ and b) 6 NH volcanic eruptions, where anomalies are calculated with respect to 10 years
prior each eruption. Grey lines indicate the average NAO index, the blue dashed lines are upper and
lower 95% c.i. The red line indicate 95% c.l. and the orange 99% c.l. that is calculated by methods of
Monte Carlo.
This can also be observed in the NAO index, derived from the 1st PC of the reconstructed SLP. In
Figure 14a and 14b, 8 EQ and 8/7 NH volcanic eruptions (see Table 1) are stacked to retrieve a
common response in the NAO. There the decadal-to-multidecadal response is quite strong after both
EQ and NH eruptions. Furthermore, what is of interest is that years with low $\delta^{18}$O (Figure 4a and 10a)

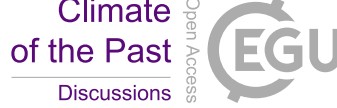

coincide with years of high NAO index (NAO+) and vice versa (Figure 14). This is particularly clear
for year 8 and 16 after NH eruptions while the first five years in the average δ¹⁸O_ice (Figure 10a) do not
show clear evidence for NAO- compared to Figure 14b.
NH eruptions seem to leave a slightly more persistent NAO response for reasons we can only speculate
at this stage. However, this does not rule out other regimes being present in these first 5 years after an
eruption, e.g. AtR after EQ eruptions or ScB after NH eruptions although the the NAO seems to be the
dominating the atmospheric circulation response. According to these results, EQ eruptions cause a clear
shift in the NAO index baseline, where a sudden increase occurs in year 1 after EQ eruptions. Although
the sudden significant decrease in year 7 and 13 is evidence of an atmospheric response it could also be
the system rebalancing. NAO index seems to oscillate more after NH eruptions. This is evident of a
weaker NAO response after NH eruptions as a result of a higher internal variability that emerges when
the NH eruptions are stacked. However, since similar trends are observed in the ice cores this suggest a
climate response forced by NH eruptions.

a)

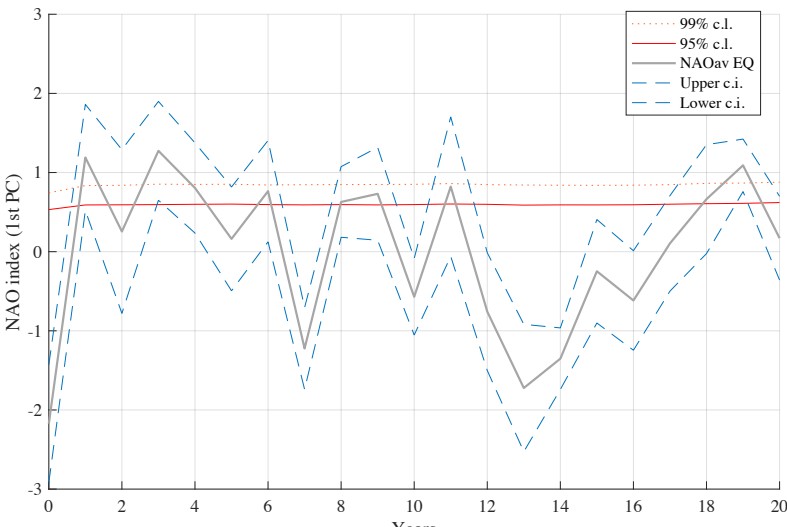






b)

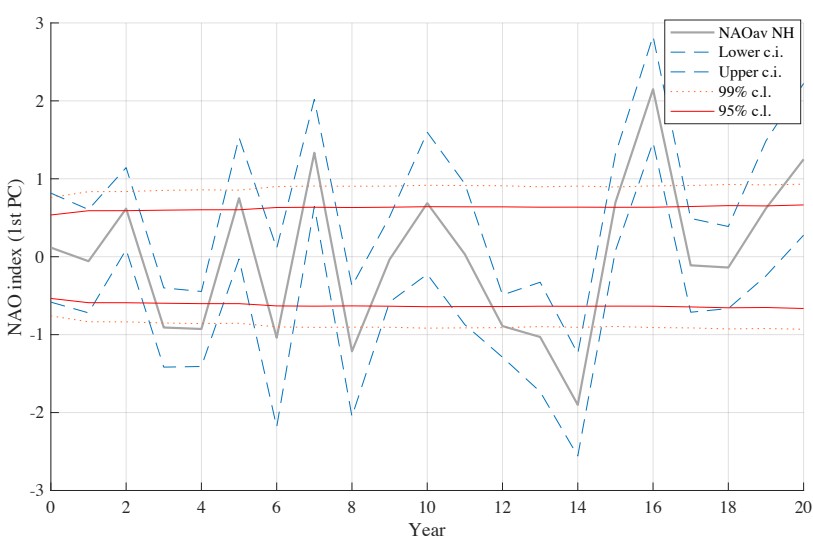


Figure 14: a) The normalized NAO index after the stacking of 8 EQ eruptions and b) the same but after
the stacking of 8 (first 5 years) and 7 (years 6-20) NH eruptions. The red and orange c.l. are calculated
by methods of Monte Carlo.

Although this study aims to complement the understanding of the atmospheric circulation response
after volcanic eruptions, as well as differentiating between the response after EQ and NH eruptions,
two important factors of the climate system are not within the scope of this study. These factors are the
ocean and sea ice, where changes in one can influence the other on a decadal to centennial time scales.
The mechanism behind the NAO- response we detect after NH eruptions may well lie in the sea ice
response to volcanic eruptions that we do not explore, especially when the results of Figure 13 are
taken into consideration. It is known that sea ice cover extent influences the planetary wave formation
by weakening the stratospheric zonal winds and therefore gives rise to NAO- (Kim et al., 2014;
Magnúsdóttir et al., 2004). This has especially been observed in relation with anthropogenic forcing,
where the decline in sea ice extent as a result of average global temperature increase has resulted in
more frequent presence of NAO- as well as ScB (Dobricic et al., 2016; Budikova, 2009 and references
therein). Therefore the understanding on the behavior of the climate system after volcanic eruptions can
serve to understand future changes in climate variability due to anthropogenic forcing and raises
questions regarding the potential positive climate feedback of NH eruptions. Furthermore, the
sensitivity of the climate system (the atmosphere, ocean and sea ice) towards NH eruptions, like e.g.
size and type of the eruption as well as the season in which the eruption occurs, remains to be assessed
in details and will be left for future studies.







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
