# Peer review of "Volcanic imprint in the North Atlantic climate"

_Climate of the Past, 2019_

## Referee Comment (RC1) · Anonymous Referee #1 · 7 Oct 2019

Summary:

The authors study the impact of volcanic eruptions on the atmospheric circulation in the North Atlantic. In order to do so they use previously published, seasonally resolved stable isotope (d18O) records from Greenland and previously published reconstructions of d18O, SLP and temperature using isotope-enabled ECHAM5-wiso GCM model simulations. The authors aim to show the different effects on atmospheric circulation (with NAO+, NAO-, Scandinavian Blocking and Atlantic Ridge as the four leading modes in the region) as imprinted in the winter d18O of the ice core records of tropical eruptions versus extra-tropical Northern hemisphere eruptions. Overall, the authors observe a

tendency that tropical eruptions are more frequently followed by a NAO+ mode (inferred from the negative d18O anomalies in the Greenland ice cores), whereas extratropical NH eruptions tend to be followed by a NAO- mode (i.e. positive anomaly in d18O). The authors further suggest that the atmospheric response persists for up to 20 years potentially involving feedbacks from ocean/atmosphere coupling.

Comments:

With this study the authors are setting out on difficult task, given the high inter-annual variability of atmospheric circulation. I am no expert in the reconstruction of weather types using stable isotopes, but I am familiar with some of the difficulties in isolating climate signals in proxy records following volcanic eruptions. Especially the criteria for (I) selecting the timespan of the analyses, (II) the climate proxy records as well as (III) the individual eruptions require further discussion and clarification.

1) Selection of the timespan / stable isotope ice-core proxies

First, a table of the stable isotope records used in your analyses would be helpful (e.g. in the SOM) so one does not need to trace them in the original publication by Vinther et al., 2010. Why does your analyses of the full timespan 1241-1978 CE only employ 3 ice cores, when there are 8 ice cores used in Sjolte et al. (2018). What are the criteria to select Dye3, GRIP and Crete? Would the outcome of your analyses be different if you used the 5 retained ice-core records? You mention that the key motivation to restrict your analyses to 1241-1978 CE is the dating issues described in Sigl et al., (2015). But this is not a valid argument, since you are basically comparing ice-core indicated d18O changes relative to ice-core indicated volcanic eruption signals. The analyses can be done independent of the absolute age accuracy of ice-core records. Volcanic reconstructions and stratigraphic age markers in Dye3, GRIP and Crete are also available on the previous chronology from Greenland (i.e., GICC05). Extending the analyses into the time period before 1241 CE would allow to include many more volcanic eruptions, especially such of large magnitudes in the NH (e.g., Katla 1179,

934, 822; Changbaishan Millennium, Churchill White River Ash; unknown eruptions in 626 and 536). Such large eruptions (with high SO2 emissions and strong negative radiative climate forcing) are currently underrepresented in your analyses. I don't know if the d18O records have enough resolution to retain a winter d18O signal before 1241 CE.

2) Selection of the volcanic eruptions

The climate impact of volcanic eruptions is primarily due to the emissions of SO2. This is well reflected in your selected tropical eruptions, but is somewhat unclear in the selected extratropical NH eruptions (Table 1). There appears to be a bias towards Icelandic eruptions, including such eruptions with no detectable sulfate in many ice cores from Greenland (i.e., Hekla 1300, Öræfajökull 1362). The VEI is a poor indicator for the climate impact potential of volcanic eruptions. A number of larger eruptions regarding their sulfate injection are missing in the list (e.g., Tarumai 1739, 1667; Fuji 1707; unidentified eruptions in 1646, 1480 or 1329. I believe the amount of sulfate deposited over Greenland available online (Sigl et al., 2013) would be a more objective criteria for selection. It would arguably also be better suited to study the climatic impacts in the North Atlantic.

3) Effects of secondary eruptions on the baseline and persistency

With roughly 300 eruptions detectable in the ice cores over the past 2500 years (Sigl et al., 2015) it is difficult to isolate the climate effects following an individual eruption, especially on decadal timescales. This is even more difficult since volcanic eruptions tend to cluster forming "double events" or "triple events" and the climate effects following such compound events are believed to be especially pronounced (Buntgen et al., 2016; Toohey et al., 2016). Isolating the effects of an individual eruption on climate and analyzing the long-term persistency – as is done in this study – requires to appropriately account for this clustering. The 10-year pre-event background period should also not be influenced by strong eruptions. In your current analyses this is often the case

(e.g. 1453 -> 1458; 1809 -> 1815; 1831 -> 1835; 1912->1918). When analyzing the long-term response up to 20 years after an eruption it is important to know that there are also many cases in which additional eruptions occured (e.g., 1258->1276 =18 yrs; 1276->1286 =10 yrs; 1458->1477 =19 yrs; 1884 -> 1902 =18 yrs; 1982 -> 1991 =9 yrs; 1963 -> 1982, 19 yrs). If these additional eruptions are not removed from the analyses it remains impossible to judge if the long term changes of d18O you discuss (centered at around 10 and 18 years) are indeed an indication of some persistency in the climate system, or simply the effects of additional volcanic events.

In summary, the current analyses provide some indications of potentially different atmospheric circulation responses following volcanic eruptions in the high-latitudes vs. eruptions from the low latitudes. The study and the robustness of the results could, however, benefit by increasing the number of volcanic eruptions in the analyses, a better tailored selection towards sulfate rich eruptions and a cleaning of the d18O records to remove the superposed effects of additional eruptions pre- and post-event.

Additional Comments:

L. 23: Typo; Atlantic Ridge

L. 40: this statement is a bit too general; also tropospheric eruptions can impact climate, e.g. when emissions are pervasive as was the case for Laki 1783, Eldgja 934, Holuhraun 2014.

L. 97: As outlined before the issues have been resolved by Sigl et al., (2015) and they are not critical for your kind of analyses (directly comparing ice-core vs. ice core).

L. 125: Typo; Extracting a volcanic signal

L. 130: Typo; extracting the long term response

L. 130: Typo; significance is estimated . . .

Table 1: Replace Eruption year with Ice Core Year (in some cases the eruption occurred

one year earlier)

Check Spelling of Krakatao, Huaynaputina and others

L. 140: Typo: another

L. 144: No! Many NH eruptions have the potential to alter the climate system (Toohey et al., 2019), there may be an absence of very large NH eruptions between 1241 and 1970; but there are many examples of strong climate impact following eruptions in the NH, the 536 AD event probably being the most prominent example

L. 145: largest in which respect? It is the SO2 amount emitted that is most important for the climate impact.

L. 152-153: VEI is not the right parameter to select eruptions for the purpose of this study

L. 157: better: North Atlantic climate response following equatorial eruptions

Figure 2: What does the stippling represent?

L. 181-82: Wouldn't one expect to find an agreement given that both reconstruction use the same d18O data?

L. 186-187: The spatial spread of ice cores appears rather limited, as you later describe. Is a positive NAO+ the only possible explanation for a negative anomaly of d18O in Central Greenland? Couldn't the low d18O values simply be the result of post-volcanic cooling, potentially prolonged by increased sea-ice formation along the Greenland coast?

L. 192-287 incl. Figs 4-6: Especially in this section it appears critical to me to discuss the potential role of secondary eruptions. You could try to remove the d18O data following secondary eruptions or stack also the volcanic forcing records so the reader can judge if the anomalies at 8-11 and 17-20 years overlap with increased volcanic activity.

L. 289: better: North Atlantic climate response following extratropical NH eruptions

L. 292: three of the five events occur during a time with already strong anthropogenic forcing (GHG, tropospheric aerosols)

L. 294: this statement is too general; the eruption year itself can have a strong climatic perturbation given the shorter lifetime of aerosols from high-latitude eruptions. It is rather a coincident that the two largest eruptions among these five have occurred in June (Laki, Katmai) so the climatic impacts were stronger in the following year.

L. 324-333: All but two (V1477 and Laki 1783) of your 7 or 8 eruptions analyzed produced comparable small sulfate deposition rates over Greenland (i.e. <10 kg km-2yr-1; Sigl et al., 2015). Almost all of them were also followed by additional eruptions 1477->1480; 1721->1729, 1739; 1755-> 1762, 1766; 1947->1956, 1963 in many cases exceeding your investigated events regarding sulfate mass injection. I am very reluctant to interpret the apparent long term changes in d18O is a long-term effect on the climate system from the original eruption. How sensitive is the outcome of the analyses from the choice of your eruptions?

L. 403-408: What are the prospects to incorporate more records from North Greenland? What are the limitations?

L. 413: Typo: Check sentence

L. 419-420: Is ECHAM5 the only model that does not produce a NAO+ after the eruption? The only one that is suggested to overestimate surface cooling? Is the surface cooling overestimated globally?

L. 421: Which reconstructions?

L. 424: I agree that more data is certainly needed; including more eruptions of higher magnitude.

L. 428-29: I haven't read their papers but I can imagine it is hard to link sea-ice vari-

ability with certainty to a mode of the NAO.

L. 433-436: It is difficult to understand the different responses of the climate system to different volcanic eruptions since there are many parameters that may have an influence. Eruption source parameters (season of the eruption, plume height, aerosol size) may be different as well as the background state of the climate system in different time windows (sea-ice, previous volcanic eruptions, other forcings).

L. 490-506: If I understand correctly you are implying that a positive NAO index leads to less precipitation over Greenland. However, you restrict your analyses to test this to the last 300 years and comparable small volcanic eruptions, leading to rel. weak observed changes in accumulation. You could easily extend this analyses to other ice cores and longer timescales. Both NGRIP and NEEM have an annual-layer counted chronology covering most of the Common Era. This would allow you to get access to a larger number of eruptions (at least about 50 events tropical and 50 NH) of larger magnitude, which should narrow your confidence intervals. So most of the needed data is already there.

L. 560: Which NAO index are you showing? Please add citation.

L. 572: Here you state that anthropogenic forcing also interplays with atmospheric circulation, yet in your previous analyses you do not exclude those eruptions occurring under strong anthropogenic forcing (20th century).

References You could include in your study a few recent papers added below (*) aiming at analyzing the effects of volcanoes and other aerosols on climate variability in the Northern Hemisphere.

*Birkel, S. D., Mayewski, P. A., Maasch, K. A., Kurbatov, A. V., and Lyon, B., 2018, Evidence for a volcanic underpinning of the Atlantic multidecadal oscillation: Npj Climate and Atmospheric Science, v. 1.

*Booth, B. B. B., Dunstone, N. J., Halloran, P. R., Andrews, T., and Bellouin, N., 2012,

Aerosols implicated as a prime driver of twentieth-century North Atlantic climate variability (vol 484, pg 228, 2012): Nature, v. 485, no. 7399, p. 534-534.

Buntgen, U., Myglan, V. S., Ljungqvist, F. C., McCormick, M., Di Cosmo, N., Sigl, M., Jungclaus, J., Wagner, S., Krusic, P. J., Esper, J., Kaplan, J. O., de Vaan, M. A. C., Luterbacher, J., Wacker, L., Tegel, W., and Kirdyanov, A. V., 2016, Cooling and societal change during the Late Antique Little Ice Age from 536 to around 660 AD: Nature Geoscience, v. 9, no. 3, p. 231-U163.

*Illing, S., Kadow, C., Pohlmann, H., and Timmreck, C., 2018, Assessing the impact of a future volcanic eruption on decadal predictions: Earth System Dynamics, v. 9, no. 2, p. 701-715.

Sigl, M., McConnell, J. R., Layman, L., Maselli, O., McGwire, K., Pasteris, D., Dahl-Jensen, D., Steffensen, J. P., Vinther, B., Edwards, R., Mulvaney, R., and Kipfstuhl, S., 2013, A new bipolar ice core record of volcanism from WAIS Divide and NEEM and implications for climate forcing of the last 2000 years: Journal of Geophysical Research-Atmospheres, v. 118, no. 3, p. 1151-1169.

Toohey, M., Kruger, K., Schmidt, H., Timmreck, C., Sigl, M., Stoffel, M., and Wilson, R., 2019, Disproportionately strong climate forcing from extratropical explosive volcanic eruptions: Nature Geoscience, v. 12, no. 2, p. 100-+.

Toohey, M., Kruger, K., Sigl, M., Stordal, F., and Svensen, H., 2016, Climatic and societal impacts of a volcanic double event at the dawn of the Middle Ages: Climatic Change, v. 136, no. 3-4, p. 401-412.

*Zanchettin, D., Timmreck, C., Toohey, M., Jungclaus, J. H., Bittner, M., Lorenz, S. J., and Rubino, A., 2019, Clarifying the Relative Role of Forcing Uncertainties and Initial-Condition Unknowns in Spreading the Climate Response to Volcanic Eruptions: Geophysical Research Letters, v. 46, no. 3, p. 1602-1611.

---

## Referee Comment (RC2) · Anonymous Referee #2 · 11 Oct 2019

———————

General comments

———————

This paper proposes to evaluate the impact of volcanic eruptions on the North Atlantic circulation using both Greenland ice core data and climate simulations of the last millennium including isotopes. The focus is put both on the response to high latitude and tropical eruptions. The analysis searches for responses up to 20 years after the onset of the eruption using climate weather regimes. The main results suggest that tropical eruptions may enhance the occurrence of NAO+ and Atlantic Ridge weather regimes

while NAO- and Scandinavian Blocking are favored by high latitude eruptions.

The topic of investigation is interesting and the questions tackled are very relevant to improve our knowledge on the response of atmospheric circulation to volcanic eruptions, which is still uncertain. The methodology sounds promising, notably the use of weather regimes. Nevertheless, I have number of serious issues that may prevent the publication of this paper in its current form.

1) The statistical analysis is poorly depicted and the use of weather regimes is not appropriately done in my opinion. The authors do find a number of significant responses, even 20 years after the eruptions, but the details concerning the way the significance is evaluated is not sufficient to be understandable. As shown by Menegoz et al. (2018), the variability in the North Atlantic sector is very large, and the need for a lot of members is absolutely necessary to have robust responses. The use of only 5 volcanic eruptions in the composite analysis may prevent any robust conclusions. More should be done concerning this statistical significance of the results using for instance bootstrap techniques. Also, the methodology related with the use of weather regimes is not clear at all. Menegoz et al. (2018) is also providing useful diagnostics to use this tool. For instance, Wiskher plots seem absolutely necessary to evaluate the impact of volcanic eruptions on the occurrence of the different regimes following the eruptions, in order to gain insights on the significance of the results, as compared to internal variability of the climate model. Once again, more than 5 members will be necessary for this. Furthermore, I am a bit surprised to find strong signal in the atmosphere so late after the onset of the eruption. As mentioned in the paper, this could be due to an an interaction with long-term changes in the ocean and sea ice, but usually such interactions are very weak in climate models.

2) There is a total lack of dynamical analysis of the results. In a sense, the analysis is restricted to repeating the same composite diagnostic to several volcanic eruptions or dataset. This is a bit short in my view to provide clear insights on what is going on following the volcanic eruptions analyzed. The use of a climate model should allow to

dig a little bit into the processes at play, which is crucial to provide an improvement in our understanding.

3) The novelty as compared to previous studies using similar models are poorly discussed and the experimental design of the simulation is not precise enough. A similar analysis by Zanchettin et al. (2012) seems to find similar results concerning a long-term response to volcanic eruptions, while Toohey et al. (2014) found that the response might be very sensitive to the way the forcing related with volcanic eruptions is implemented. Also, the two former studies used different statistical analysis of the significance of their signal, which is indeed crucial. What is this new study providing in regard of these two former studies using the same family of climate models?

You will find in the following a few more detailed suggestions that may allow to improve the manuscript and bring it towards publication standard of climate of the past.

——————————

Specific comments

——————————

- Introduction: please provide a few references to substantiate it (e.g. review by Robock 2000, Timmreck et al. 2012; Swingedouw et al. 2017)

- l. 49: You jump from 1st EOF to third. What is the second? Usually it is the East Atlantic Pattern, a mode whose negative phase resembles the Atlantic Ridge. Please read more carefully Cassou et al. (2004) who is describing this (they mentioned this as well).

- l. 83: replace "compliment" by "complement"

- l. 106: since the model simulation is only assimilating data from ice cores, it is not a big surprise that the results from ice core data and model agree.

- l. 112-114: This sentence is not clear at all, nor the following one. I do not get

what is done. Also, you should specify which type of data you used for the clustering. Usually weather regimes are made using daily data. Is it what have been done? Also, it is not mentioned which season is used. All this part concerning the weather regime computation using clustering technique should be largely improved. Also, I do not get what is done with this decomposition since it is not much discussed afterwards in the manuscript which is mainly showing composite of O18, temperature or SLP. Please clarify your approach here and do a proper analysis of weather regime changes in occurrence as done in Menegoz et al. (2018) for instance.

- L. 121: what is "r"? Correlation I assume. Here I do not get what is done.

- L. 132: Your Monte Carlo approach should be further described

- L. 135: If I understand correctly you are doing a Student t-test to evaluate the significance of the composite, with a degree of freedom of 5, equal to the number "n" of volcanic eruptions? Usually it is n-1 that is used for the degrees of freedom in a Student t-test. Thus, you should end up with 4 (this may change the significance of your results), which is very small, once again highlighting the need for more members in your analysis of the very noisy atmospheric circulation. Bootstrap approaches may be more appropriate here to evaluate the significance of your composite, since it will better account for the fact that the atmospheric variables are usually not independent variable. Analysis of significance is very key for your results, so a very particular care should be taken here to avoid to analyze noise-induced signals (which I suspect given the long-term response found, which is quite unusual in the atmosphere that no memory). In that sense, the analysis of processes at play should also help to evaluate if the signal you find is coherent in terms of dynamics.

- L. 141-142: the radiative forcing estimate are coming from Sigl et al. (2015) according to Table 1. Nevertheless, the simulation you are analyzing is using Jungclaus et al. (2010) forcing, so I assume a different forcing than Sigl et al. (2015). Even though the latter is certainly better than the former, since it is more up to date, I think you need

to be coherent with the forcing used in the simulation. At least an evaluation of the differences concerning the volcanic eruption selected should be done.

- L. 146: a reference would be useful to support this claim.

- L. 185-186: "when compared with O18 pattern..." I do not get why the clustering is not use here to estimate the changes in occurrence in the different regimes. This would help to better evaluate the significance in the results and would be more coherent than a spatial correlation. Indeed, as shown in Toohey et al. (2014) the response to volcanic eruptions might be different than the preferential modes of variability. Also, this diagnostic of spatial correlation is dubious, since several patterns can be correlated, while the counting of day within each regime is far more instructive quantitatively.

- L. 415: "already reported by Sjolte et al. (2018)" then you should clearly highlight what is new in your study as compared to this former one that also look at the impact in volcanic eruptions in the same simulation...

- L. 424:" identification of this prolonged NAO+ in O18". You only found a NAO-like response in your O18 data, not a prolonged NAO+ (your data is not NAO...). Furthermore, such a signal not present in Ortega et al. NAO reconstruction... Is it in other NAO reconstructions?

- L. 430: a mechanism is mentioned, but nothing to substantiate it. Will it be possible to dig a bit more towards mechanistic understanding of your signal (and a better synthesis of your results to gain space, or just show the most robust and new ones).

- L. 440: Can you remind the reader how Zanchettin et al. (2012) explained their results. Since you use a similar model, can you check that the same mechanism is at play?

- L. 443: Can you explain how you relate the changes in the AMOC with your NAO response. See Gastineau et al. (2011) for some hints.

- Conclusion: since your study is mainly based on ice core data (either directly from

Greenland data or assimilation run that only assimilate this data) this should be high-lighted as a major caveat. Indeed, even though the variability mode has different impacts over Greenland, Greenland is covering only a small part of the weather regime fingerprints, and the use of over regions might be very useful (cf. Ortega et al. 2015).

- L. 477: I do think you need to try answer this. Indeed, when you will have sorted out the really robust signal and novelty as compared to former studies using same data, you may need to provide mechanistic interpretation of the results, this what model simulations are made for in a sense: understanding processes to corroborate hypotheses.

————————

Additional references:

————————

Gastineau G, Frankignoul C (2011) Cold-season atmospheric response to the natural variability of the Atlantic meridional overturning circulation. Clim Dyn. doi:10.1007/s00382-011-1109-y

Menegoz M., Cassou C., Swingedouw D., Bretonniere P.-A., Doblas-Reyes P. (2018) Modulation of the climate response to a volcanic eruption by the Atlantic Multidecadal Variability. Climate dynamics 51 (5-6), pp. 1863-1883.

Robock, A., 2000. Volcanic eruptions and climate. Rev. Geophys. 38 (2), 191–219.

Swingedouw D., Mignot J., Ortega P., Khodri M., Menegoz M., Cassou C. and Hanquiez V. (2017) Impact of explosive volcanic eruptions on the main climate variability modes. Global and Planetary Changes 150, pp. 24-45.

Timmreck, C., 2012. Modeling the climatic effects of large explosive volcanic eruptions. Wiley Interdiscip. Rev. Clim. Chang. 3 (6), 545–564.

Toohey, M., et al., 2014. The Impact of Volcanic Aerosol on the Northern Hemisphere Stratospheric Polar Vortex: Mechanisms and Sensitivity to Forcing Structure.

pp. 16777–16819.

---

## Editor Comment (EC1) · Eric Wolff (Editor) · 18 Oct 2019

Your paper has received two reviews. The discussion will still be open for a couple of weeks and it is possible (though unlikely) that there could be further comments. As you know, you are expected to respond to each of the review comments, and after that I have to give a formal editorial decision. However I think it will be useful if I give you an idea of where we are with your paper now, as it may help you to think about how best to respond to the reviewers.

The reviews, although not going as far as to recommend rejection of your paper, are quite negative, seeing substantial problems with the statistical significance of your re-

sults, both because of the number of ice cores, the number of eruptions, and the statistical methods used to determine significance. Reviewer 1 also makes an extremely important point about the validity of looking at times after an eruption without considering secondary eruptions and "cleaning" for them.

For the discussion of statistical significance, I add another issue about the presentation of your data. Readers naturally wish to compare the ice core data in eg Fig 1 with the modelled 18O output in Fig 2 (and the same for subsequent figures). However the colour scheme you have selected in Fig 2 and subsequent figures makes this impossible. I have tried using the online version in case it was a problem with the way the figures printed but it makes no difference: it's just impossible to see any significant effect of the eruptions in Fig 2 and most following figures because the contours are almost all the same colour. You need to think hard about how to present this more effectively: a zoom on Greenland might help; using solid shading rather than feint contour lines might help; stretching the colour scale more might help. A further problem exists with Figure 1, because I cannot see what is said to be significant: the very few circles with a red edging (95%) are clear, but I see nothing with an obviously magenta edge (do you mean what looks like black?). But anyway as it stands at present when you say (line 181) "A general agreement can be found" (between Fig 1 and 2), I just can't see it. In fact to be honest, it looks as if its true that there is a general agreement that the eruptions had little statistically significant effect on 18O.

Taking into account the comments of both reviewers and my own misgivings about the presentation of significance, I want to emphasise that a revision that only makes cosmetic changes to the paper will be rejected. In order to convince the reviewers you will need in your response to state how you will very strongly improve the evidence that the data are in support of the modelled findings. This might involve looking at more large eruptions as suggested by rev 1, or by a much improved statistical treatment and presentation. I would like to see what you plan in your response before I would be willing to recommend preparing a new version.

---

## Author Comment (AC1) · 24 Oct 2019

Dear editor,

We are thankful for your guidance concerning the next steps in our response. We are also grateful for the reviewer's comments, and taking them into account will certainly improve our study. Looking at the reviewer's comments, it is obvious to us that a revision of the text as well as more analysis is needed to support our study.

We propose to conduct the following analysis and tests:

Extend the analysis of stable isotope data further back and use more volcanic erup-

tions. Although our decision to use the period of 1241-1979 within the ice cores is to have the results comparable to the results for the atmospheric circulation reconstruction, which covers the period1241-1970, it will be beneficial to look at more eruptions as an assessment of our results. In the manuscript we will furthermore explain the reason for choosing the reference period better since it seems to cause confusion.

Perform sensitivity tests including and excluding eruptions not found in Sigl et al. 2015, but documented in other sources, such as historical data, other climate proxies like sediment cores and/or Zielinski et al., 1994, to test the dependency of the choice of eruptions. Our argument is that the signal detected in Greenland ice cores is dependent on wind direction, dispersion and deposition of the plume at the time of, and after, the eruption. If the volcanic eruption is not detected as a sulfate peak in the core itself it could still have an impact on the atmospheric circulation (both short and long-term impact) that can be identified in the ice cores as stable isotope anomalies.

Use bootstrapping method and compare with our results using Monte Carlo method, to further test the significance of the isotopic response to volcanic eruptions.

Re-assess the pre-eruption baseline used.

Re-assess figure graphics and use of color schemes to make the figures easier to read.

There are also several important factors that were not clear enough in the manuscript that have caused confusion and need to be clarified. This is e.g. that we are not using model runs, we use a model-data atmospheric circulation reconstruction to compare with the ice core data. Reviewer #2 seems to have misunderstood this important point. We will carefully reply to all reviewer comments and go through the manuscript to clarify the writing.

---

## Author Comment (AC2) · 7 Jan 2020

Dear reviewer,

We are thankful for your instructive comments that will indeed serve as an improvement to our study. It is clear that several factors needs improvements. Other factors seem to cause confusion that need further clarifications. We have provided some answers/clarifications to the reviewer's comments here below: 1) Selection of the timespan/stable isotope ice-core proxies: The time span of this study is mainly selected so the Greenland ice cores's results can be compared to the reconstruction from Sjolte et al. (2018) Our motivation is to be able to identify the imprint of volcanic eruptions in

individual ice cores (Dye-3, GRIP and Crete). They do have stable isotope data further back in time (back to 551 AD) compared to the reconstruction by Sjolte et al (2018) and it is true that it is well possible to add more eruptions for the ice core analysis. This we have done and a comparison of the results between the timespan of 1241-1978 and 551-1978 will be added into the manuscript. 2) Selection of the volcanic eruptions: We agree on what is said here. It is difficult to select NH eruptions since they are a) quite frequent and b) their signal can be influenced by EQ eruptions. Therefore we have extended the time period for the Greenland ice core analysis to 551 (as stated above) and added more NH eruptions (in total eight eruptions). Some of the eruptions mentioned by reviewer do have a mixed signal due to other NH eruption occuring close in time. As a result, the number of NH eruptions used for the short-term climate signal using ice cores spanning the time period 1771-1970 will decrease (1918 eruption removed) to four. Therefore more robust statistical methods will be used to evaluate the significance of the signal identified. 3) Effects of secondary eruptions on the baseline and persistency: Due to the re-assessment done in our respones to comment Nr. 1 and 2, this has also been taken into consideration and relevant eruptions removed from the analysis to form a more solid baseline.

In addition here below are the authors response (AR) to the additional comments: L. 23: Typo; Atlantic Ridge: AR: Ok.

L. 40: this statement is a bit too general; also tropospheric eruptions can impact climate, e.g. when emissions are pervasive as was the case for Laki 1783, Eldgja 934,Holuhraun 2014.: AR: Ok, this will be changed. However, Laki is considered to have been a mixed eruption (both tropospheric and stratospheric) (Thordarson and Self, 2003) (Ref: Thordarson, T., & Self, S. (2003). Atmospheric and environmental effects of the 1783–1784 Laki eruption: A review and reassessment. Journal of Geophysical Research: Atmospheres, 108(D1), AAC-7.)

L. 97: As outlined before the issues have been resolved by Sigl et al., (2015) and they are not critical for your kind of analyses (directly comparing ice-core vs. ice core). AR:

[Figure]

Ok, we will clarify this in a revised manuscript.

L. 125: Typo; Extracting a volcanic signal. AR: Ok.

L. 130: Typo; extracting the long term response. AR: Ok.

L. 130: Typo; significance is estimated.. AR: Ok.

Table 1: Replace Eruption year with Ice Core Year (in some cases the eruption occurred one year earlier). AR: Ok.

Check Spelling of Krakatao, Huaynaputina and others. AR: Ok.

L. 140: Typo: another. AR: Ok.

L. 144: No! Many NH eruptions have the potential to alter the climate system (Toohey et al., 2019), there may be an absence of very large NH eruptions between 1241 and1970; but there are many examples of strong climate impact following eruptions in the NH, the 536 AD event probably being the most prominent example. AR: We agree and this sentence does not reflect what we meant to say. This will be clarified.

L. 145: largest in which respect? It is the SO2 amount emitted that is most important for the climate impact. AR: Indeed that is true and we will rephrase this sentence.

L. 152-153: VEI is not the right parameter to select eruptions for the purpose of this study. AR: Ok.

L. 157: better: North Atlantic climate response following equatorial eruptions. AR: Ok.

Figure 2: What does the stippling represent? AR: Here it represents standardized SLP values, where the mean=0 and standard deviation=1. We will add this in a revised manuscript where appropriate.

L. 181-82: Wouldn't one expect to find an agreement given that both reconstruction use the same d18O data? AR: In principle, yes, but in this case we are comparing the raw data of 13 ice cores to a reconstruction based on fitting modeled d18O with

8 ice cores. There are statistical and methodological differences, which high light the challenges in detecting the volcanic response in individual ice cores. We have clarified this in the text.

L. 186-187: The spatial spread of ice cores appears rather limited, as you later describe. Is a positive NAO+ the only possible explanation for a negative anomaly ofd18O in Central Greenland? Couldn't the low d18O values simply be the result of post-volcanic cooling, potentially prolonged by increased sea-ice formation along the Greenland coast? AR: This is indeed a possibility, but then the spatial pattern could indicate the origin of the center of the cooling. Especially after equatorial eruptions, the increase in arctic sea ice extent would suggest a gradient with negative anomalies from west-east or north-south (or there between). We will assess this further and add a note on this if appropriate.

L. 192-287 incl. Figs 4-6: Especially in this section it appears critical to me to discuss the potential role of secondary eruptions. You could try to remove the d18O data following secondary eruptions or stack also the volcanic forcing records so the reader can judge if the anomalies at 8-11 and 17-20 years overlap with increased volcanic activity. AR: As a result of the re-evaluation in relation to the main comments, this will also be re-assessed in a revised version.

L. 289: better: North Atlantic climate response following extratropical NH eruptions. AR: Ok.

L. 292: three of the five events occur during a time with already strong anthropogenic forcing (GHG, tropospheric aerosols). AR: Ok, will consider.

L. 294: this statement is too general; the eruption year itself can have a strong climatic perturbation given the shorter lifetime of aerosols from high-latitude eruptions. It is rather a coincident that the two largest eruptions among these five have occurred in June (Laki, Katmai) so the climatic impacts were stronger in the following year. AR: Ok, will rephrase sentence/paragraph in a revised version.

L. 324-333: All but two (V1477 and Laki 1783) of your 7 or 8 eruptions analyzed produced comparable small sulfate deposition rates over Greenland (i.e. <10 kg km-2yr-1;Sigl et al., 2015). Almost all of them were also followed by additional eruptions 1477->1480; 1721->1729, 1739; 1755-> 1762, 1766; 1947->1956, 1963 in many cases exceeding your investigated events regarding sulfate mass injection. I am very reluctant to interpret the apparent long term changes in d18O is a long-term effect on the climate system from the original eruption. How sensitive is the outcome of the analyses from the choice of your eruptions? AR: Again, as a result of our re-assessment to the main comments, this will be re-evaluated in a revised version.

L. 403-408: What are the prospects to incorporate more records from North Greenland? What are the limitations? AR: In this study we focus on the winter signal. The climate becomes more continental going further north, which mean less winter precipitation and a degradation of the signal to noise ratio in the records (e.g. Zheng et al., 2018) (Ref: https://www.clim-past.net/14/1067/2018/). Accumulation also decreases which makes it difficult to retrieve the seasonal signal due to diffusion obliterating the d18O annual cycle

L. 413: Typo: Check sentence. AR: Ok.

L. 419-420: Is ECHAM5 the only model that does not produce a NAO+ after the eruption? The only one that is suggested to overestimate surface cooling? Is the surface cooling overestimated globally? AR: No, the CMIP5 models generally tend to overestimate surface cooling after volcanic eruptions and the dynamic NH climate response following equatorial eruptions. This will be clarified.

L. 421: Which reconstructions? AR: The reconstructions of Sjolte et al. (2018). This will be clarified in text in a revised manuscript.

L. 424: I agree that more data is certainly needed; including more eruptions of higher magnitude. AR: Ok.

[Figure]

L. 428-29: I haven't read their papers but I can imagine it is hard to link sea-ice variability with certainty to a mode of the NAO. AR: Perhaps it is hard to imagine, but their work/argument is solid and robust and is ongoing within this subject.

L. 433-436: It is difficult to understand the different responses of the climate system to different volcanic eruptions since there are many parameters that may have an influence. Eruption source parameters (season of the eruption, plume height, aerosol size) may be different as well as the background state of the climate system in different time windows (sea-ice, previous volcanic eruptions, other forcings). AR: Indeed, this is an ongoing research subject with many unanswered questions.

L. 490-506: If I understand correctly you are implying that a positive NAO index leads to less precipitation over Greenland. However, you restrict your analyses to test this to the last 300 years and comparable small volcanic eruptions, leading to rel. weak observed changes in accumulation. You could easily extend this analyses to other ice cores and longer timescales. Both NGRIP and NEEM have an annual-layer counted chronology covering most of the Common Era. This would allow you to get access toa larger number of eruptions (at least about 50 events tropical and 50 NH) of larger magnitude, which should narrow your confidence intervals. So most of the needed data is already there. AR: We see your point and we will assess if additional data will be added to support our hypothesis in a revised manuscript.

L. 560: Which NAO index are you showing? Please add citation. The one we have calculated using the reconstructions of Sjolte et al. (2018). AR: We will underline this better since this is not clear.

L. 572: Here you state that anthropogenic forcing also interplays with atmospheric circulation, yet in your previous analyses you do not exclude those eruptions occurring under strong anthropogenic forcing (20th century). AR: We thank the reviewer for his excellent precision and we will assess if a comparison will be added in a revised version.

References You could include in your study a few recent papers added below (*) aiming at analyzing the effects of volcanoes and other aerosols on climate variability in the Northern Hemisphere. AR: We thank the reviewer for these relevant reference suggestions that will be added.

---

## Author Comment (AC3) · 7 Jan 2020

Dear reviewer, We are thankful for your instructive comments that will indeed serve as an improvement to our study. It is clear that several factors needs improvements. Other factors seem to cause confusion that need further clarifications. We have provided some answers/clarifications to the reviewer's comments here below:

1) First issue: We do see the reviewer's point and will address this in a revised manuscript where the bootstrapping method will be used as well as assess the use of other methods (e.g. Whisker plots). Furthermore, as can be seen in the answer's to reviewer #1, more eruptions have been added to assess the long-term climate signal

in the Greenland ice cores (dates back to 551 AD). This will however be difficult for the time period of 1241-1978 due to the frequency of NH eruptions and the disturbance of EQ eruptions on the climate signal of NH eruptions. This will be discussed in detail and a clear comparison between five and eight eruption results made.

2) Second issue: This is true. However, this study is not a modelling study and therefore we cannot use the model, that forms the basis for the atmospheric circulation reconstructions used (by Sjolte et al. (2018)), to do further experiments. We are merely using the selected reconstruction parameters to assess the signal identified in the stable water isotopes of Greenland ice cores and assign the climate signal to a specific weather regime if possible. This important point needs clarification that will be added to a revised manuscript.

3) Third issue: We will add more precise discussions on previous model studies that are relevant to ours and clarify the novelty of ours.

In addition here below are the authors response (AR) to the additional comments:

Introduction: please provide a few references to substantiate it (e.g. review by Robock2000, Timmreck et al. 2012; Swingedouw et al. 2017). AR: Ok.

l. 49: You jump from 1st EOF to third. What is the second? Usually it is the East Atlantic Pattern, a mode whose negative phase resembles the Atlantic Ridge. Please read more carefully Cassou et al. (2004) who is describing this (they mentioned this as well). AR: Yes, we are aware and will clarify this.

l. 83: replace "compliment" by "complement". AR: Ok.

l. 106: since the model simulation is only assimilating data from ice cores, it is not a big surprise that the results from ice core data and model agree. AR: That is true. However, since we are studying if specific weather regimes are present after volcanic eruptions and if they can be identified in Greenland ice cores by using the reconstructions as a reference (and different method), this needs testing.

l. 112-114: This sentence is not clear at all, nor the following one. I do not get what is done. Also, you should specify which type of data you used for the clustering. Usually weather regimes are made using daily data. Is it what have been done? Also, it is not mentioned which season is used. All this part concerning the weather regime computation using clustering technique should be largely improved. Also, I do not get what is done with this decomposition since it is not much discussed afterwards in the manuscript which is mainly showing composite of O18, temperature or SLP. Please clarify your approach here and do a proper analysis of weather regime changes in occurrence as done in Menegoz et al. (2018) for instance. AR: We thank the reviewer for his detailed eye and indeed an improvement is needed. Monthly mean standardized data is used - and to compare with Greenland ice cores we only use winter data/output. This section will be improved and explained in more detail in a revised version.

L. 121: what is "r"? Correlation I assume. Here I do not get what is done. AR: Yes, correlation. In order to assign a post volcanic atmospheric circulation field (retrieved from the reconstructions) to one of the four main weather regimes in the NA with some certainty, say in year 2, we compare the post-volcanic field to the 1200-year average weather regimes retrieved by clustering ECHAM5 SLP by calculating the correlation coefficient r. This will be further clarified in a revised manuscript.

L. 132: Your Monte Carlo approach should be further described. AR: Ok.

L. 135: If I understand correctly you are doing a Student t-test to evaluate the significance of the composite, with a degree of freedom of 5, equal to the number "n" of volcanic eruptions? Usually it is n-1 that is used for the degrees of freedom in a Student t-test. Thus, you should end up with 4 (this may change the significance of your results), which is very small, once again highlighting the need for more members in your analysis of the very noisy atmospheric circulation. Bootstrap approaches maybe more appropriate here to evaluate the significance of your composite, since it will better account for the fact that the atmospheric variables are usually not independent variable. Analysis of significance is very key for your results, so a very particular care should be

taken here to avoid to analyze noise-induced signals (which I suspect given the long-term response found, which is quite unusual in the atmosphere that no memory). In that sense, the analysis of processes at play should also help to evaluate if the signal you find is coherent in terms of dynamics. AR: As mentioned in the reply to the main comments above, this will be improved in a revised version.

L. 141-142: the radiative forcing estimate are coming from Sigl et al. (2015) according to Table 1. Nevertheless, the simulation you are analyzing is using Jungclaus et al.(2010) forcing, so I assume a different forcing than Sigl et al. (2015). Even though the latter is certainly better than the former, since it is more up to date, I think you need to be coherent with the forcing used in the simulation. At least an evaluation of the differences concerning the volcanic eruption selected should be done. AR: This we will do and add a note on this in the manuscript.

L. 146: a reference would be useful to support this claim. AR: Ok.

L. 185-186: "when compared with O18 pattern..." I do not get why the clustering is not use here to estimate the changes in occurrence in the different regimes. This would help to better evaluate the significance in the results and would be more coherent than a spatial correlation. Indeed, as shown in Toohey et al. (2014) the response to volcanic eruptions might be different than the preferential modes of variability. Also, this diagnostic of spatial correlation is dubious, since several patterns can be correlated, while the counting of day within each regime is far more instructive quantitatively. AR: Indeed that is true and it would be ideal if possible. However, the four weather regimes do not emerge in the clustering of the reconstructed d18O, due to the variability of the reconstruction being biased towards NAO-type variability as well as weather patterns and isotope patterns not being a one-to-one match.

L. 415: "already reported by Sjolte et al. (2018)" then you should clearly highlight what is new in your study as compared to this former one that also look at the impact in volcanic eruptions in the same simulation.. AR: Indeed that is true, but we would also

like to emphasize that we are not using a simulation in our study but an atmospheric circulation reconstructions (along with Greenland ice cores). Since this seems to cause some confusion we will clarify this in a revised version.

L. 424:" identification of this prolonged NAO+ in O18". You only found a NAO-like response in your O18 data, not a prolonged NAO+ (your data is not NAO...). Furthermore, such a signal not present in Ortega et al. NAO reconstruction...Is it in other NAO reconstructions? AR: That is true, we will rephase and look at the reconstruction statement more closely.

L. 430: a mechanism is mentioned, but nothing to substantiate it. Will it be possible to dig a bit more towards mechanistic understanding of your signal (and a better synthesis of your results to gain space, or just show the most robust and new ones). AR: Here we might have stepped further than reasonable concerning the fact that we are not doing model experiments. This will be altered for to avoid over-interpretations.

L. 440: Can you remind the reader how Zanchettin et al. (2012) explained their results. Since you use a similar model, can you check that the same mechanism is at play? AR: Indeed we can, an excellent point that will be added in a revised version.

L. 443: Can you explain how you relate the changes in the AMOC with your NAO response. See Gastineau et al. (2011) for some hints. AR: Again, a good suggestion and will be added in a revised version. Conclusion: since your study is mainly based on ice core data (either directly from Greenland data or assimilation run that only assimilate this data) this should be high-lighted as a major caveat. Indeed, even though the variability mode has different impacts over Greenland, Greenland is covering only a small part of the weather regime fingerprints, and the use of over regions might be very useful (cf. Ortega et al. 2015). AR: Yes, we agree and will add such a note in a revised manuscript.

L. 477: I do think you need to try answer this. Indeed, when you will have sorted out the really robust signal and novelty as compared to former studies using same data,

you may need to provide mechanistic interpretation of the results, this what model simulations are made for in a sense: understanding processes to corroborate hypotheses. AR: We do agree and aim to do exactly that in our future research. However, since this is particular study is not a modelling study this will be difficult except suggesting possible mechanism.